# KNOWLEDGE-CENTRIC DATA SELECTION FOR EFFECTIVE DOMAIN ADAPTATION OF LARGE LANGUAGE MODELS

## ABSTRACT

Domain adaptation of language models is critical for specialized applications in fields, but its success hinges on high-quality data selection rather than sheer volume. Current methods, such as heuristic filters, perplexity pruning, and embedding-based clustering, often fail to address domain-specific redundancy and noisy or overlapping data. As a result, training becomes inefficient and resource-intensive, and models may overfit to frequent linguistic patterns rather than capturing core knowledge. The resulting data-induced inefficiency limits model generalization and translates directly into prohibitive curation and computational costs. For high-stakes domains, this inefficiency is particularly detrimental, as even minor errors carry significant consequences.. We propose a knowledge-centric approach that redefines data quality around discrete knowledge procedures and theorems. Our framework introduces Knowledge Coverage Entropy (KCE), a metric quantifying knowledge diversity, and Entropy-Driven Selection (EDS), which optimizes data selection for compact, high-quality datasets. Experiments in supervised fine-tuning (SFT) and retrieval-augmented generation (RAG) demonstrate EDS's efficacy. In SFT on the MATH-500 benchmark, at matched data budgets, our method consistently yields the best post-training accuracy among data selection methods. In RAG on medical datasets, our method delivers the best retrieval quality with mean reciprocal rank (MRR) improvements of approximately 11% to 42% and substantial coverage gains while using significantly fewer samples. Enhanced performance in both SFT and RAG demonstrates that KCE offers a principled metric for data quality, and that EDS facilitates efficient in domain-specific tasks.

## 1 INTRODUCTION

Domain adaptation tailors general-purpose language models for specialized tasks, embedding domain-specific knowledge and reasoning (Howard & Ruder, 2018; Longpre et al., 2023; Seto et al., 2025; Parmar et al., 2024). Unlike broad fluency training, adaptation via supervised fine-tuning (SFT) or retrieval-augmented generation (RAG) prioritizes precise, context-relevant concepts (Shum et al., 2024; Muennighoff et al., 2025a). Effective adaptation hinges on data quality, not volume, requiring corpora that capture essential knowledge for model internalization (Pang et al., 2025; Xia et al., 2024b; Liu et al., 2023). Uncurated datasets yield diminishing returns or degraded performance under minimum description length principles (Li & Vitányi, 2008), emphasizing the need for high-quality selection to optimize learning, efficiency, and robustness (Longpre et al., 2023; Seto et al., 2025).

Current methods, including heuristic filters (e.g., text length, readability) (Xia et al., 2024b; Liu et al., 2023), perplexity-based pruning (Pang et al., 2025; Ankner et al., 2024), model loss–based filtering (IFD) (Li et al., 2024b), per-example gradients (LESS) (Xia et al., 2024a), embedding-based clustering (Xie et al., 2024), and entropy-driven approaches (Song et al., 2012; Lairez, 2022), predominantly operate at the token or embedding level. These approaches manage large corpora but struggle to identify domain-specific knowledge redundancy, such as rephrased definitions or overlapping evidence (Lee et al., 2022; Hei et al., 2024). Because they cannot reliably distinguish genuine novel knowledge from mere stylistic or lexical variations, they often resort to unscalable manual

curation (Liu et al., 2024; Wang et al., 2024). Consequently, SFT models memorize frequent surface patterns and falter on edge cases, while RAG systems retrieve irrelevant or redundant passages, increasing computational cost and harming generalization (Amiraz et al., 2025; Hager et al., 2024; Fayyaz et al., 2025). In high-stakes domains such as medicine or law, these token-level limitations translate directly into elevated risks.

Empirical studies underscore these challenges. Noisy SFT datasets containing incorrect or misaligned pairs degrade model accuracy and introduce biases (Liu et al., 2024; Wang et al., 2024), while unfiltered RAG datasets reduce retrieval precision (Amiraz et al., 2025). Unsupervised curricula also fail to address conceptual overlap without proper validation (Ankner et al., 2024; Pang et al., 2025). Current methods, whether heuristic, perplexity-based, loss-based, gradient-based, or embedding-clustering operate solely at the token, sequence, or embedding level. Consequently, they are blind to semantic equivalence across surface variations: different expressions of the same element are often discarded, rephrased definitions, logically equivalent proofs, or clinically identical guidelines are not recognized as redundant. This fundamental limitation leads to knowledge-level redundancy, poor coverage of rare but critical concepts, and the well-documented degradation in both SFT generalization and RAG retrieval precision.

To bridge these gaps, we introduce a knowledge-centric paradigm that operates on discrete, auditable knowledge units (e.g., mathematical theorems, clinical guidelines, legal principles) rather than tokens or continuous embeddings. Instead of approximating importance through proxies (perplexity, loss, or gradient norms), our framework constructs a binary knowledge coverage matrix and do greedy via Knowledge Coverage Entropy (KCE) and Entropy-Driven Selection (EDS) algorithm. Our approach shifting from surface-level statistics to knowledge-level accounting, directly optimizes the balance of key knowledge and maximizes novel knowledge.

In this data selection framework, KCE quantifies diversity and balance over discrete knowledge units, and EDS prioritizes novel, high-information samples to reduce redundancy. By leveraging entropy to emphasize informative coverage, the framework strengthens supervised fine-tuning learning signals and improves retrieval-augmented generation retrieval precision. On the MATH-500 benchmark, at matched data budgets, KCE-selected data yields the best post-training accuracy among data selection methods and reaches 456/500 with substantially fewer samples. In medical retrieval-augmented generation, the framework delivers the best retrieval quality with mean reciprocal rank improvements of approximately 11% to 42% alongside large coverage gains under significant data reduction. These results establish KCE and EDS as principled tools for efficient and high-performance domain adaptation.

## 2 METHODOLOGY

The Entropy-Driven Selection (EDS) methodology selects a diverse and informative subset of data samples by maximizing Knowledge Coverage Entropy (KCE) within a binary information-knowledge matrix. This approach constructs a matrix representing knowledge points across samples, computes entropy-based scores to quantify diversity, and employs a set-aware lazy-greedy algorithm to optimize subset selection under cardinality constraints.

### 2.1 BINARY INFORMATION-KNOWLEDGE MATRIX

We construct a knowledge set $\mathcal{K}$ of domain-relevant concepts and map each data sample to a binary vector over $\mathcal{K}$, forming a matrix $\mathbf{B} \in \{0,1\}^{n \times m}$, where $n$ is the number of samples, $m = |\mathcal{K}|$ is the number of knowledge points, and $\mathbf{B}_{i,j} = 1$ if sample $i$ covers knowledge point $j$, and 0 otherwise. The matrix $\mathbf{B}$ is built using Qwen-max-0125 (Team, 2025) with task-specific prompts to extract and tag concepts, as detailed in Appendix C. Only knowledge points that appear at least $n = 50$ times in the dataset are included, and ablation study on the knowledge-point matrix is presented in Appendix B.1. This matrix underpins the computation of Knowledge Coverage Entropy (KCE).

### 2.2 COVERAGE PROBABILITY DEFINITIONS

For the matrix $\mathbf{B} \in \{0,1\}^{n \times m}$, we define the smoothed coverage probability for sample $a$ as $P_a = \frac{\sum_{j=1}^{m} \mathbf{B}_{a,j} + \alpha}{m + \alpha m}$, where $\alpha = 10^{-6}$ ensures numerical stability. The joint probability distribution is

computed as $P_{i,j} = \frac{\mathbf{B}_{i,j}+\alpha/(nm)}{\sum_{i=1}^{n}\sum_{j=1}^{m}(\mathbf{B}_{i,j}+\alpha/(nm))}$. These probabilities support entropy calculations, with further details in Appendix B.2.

## 2.3 KNOWLEDGE COVERAGE ENTROPY (KCE)

For a subset $S \subseteq \{1, \ldots, n\}$ of size $|S| = h$, KCE is defined as

$$H(S) = -\sum_{j=1}^{m} p_j \log_2 p_j, \qquad p_j = \frac{1}{h}\sum_{a \in S} \mathbf{B}_{a,j},$$

where $p_j$ denotes the average coverage of knowledge point $j$ within $S$. To enable consistent comparison across subsets of different sizes, the entropy is normalized as

$$H_n(S) = \frac{H(S)}{h}.$$

To approximate the integral defined over a potentially infinite-dimensional knowledge space, we employ Monte Carlo sampling by drawing a finite number of points from the base measure. Further theoretical results for the infinite-dimensional case, together with corresponding upper bounds, are detailed in Appendix B.2.

Although this set-based formulation captures knowledge diversity effectively, its computation becomes costly when $n$ is large due to the dependence on subset interactions. To improve scalability, we introduce a computationally efficient approximation that assigns each sample an independent, single-pass score.

Let $\mathbf{B} \in \{0,1\}^{n \times m}$ be the binary information–knowledge matrix, where $\mathbf{B}_{a,j} = 1$ indicates that sample $a$ covers knowledge point $j$. The row-wise coverage probability for sample $a$ is defined as

$$P_a = \frac{1}{m}\sum_{j=1}^{m} \mathbf{B}_{a,j},$$

with corresponding entropy

$$H(a) = -P_a \log_2 P_a.$$

To incorporate knowledge importance, a weight vector $\mathbf{k} \in \mathbb{R}^m$ assigns importance $k_i$ to knowledge point $i$. The resulting scoring function for sample $a$ is

$$\text{Score}(a) = H(a) \cdot \left(1 + \gamma \sum_{i=1}^{m} k_i \mathbf{B}_{a,i}\right),$$

where $\gamma$ controls the strength of knowledge-aware weighting. The top-$s$ samples ranked by this score are selected.

This single-pass approach achieves linear-time complexity and scales efficiently to large datasets. However, its independence assumption ignores set-level interactions; therefore, it does not inherit the submodular guarantees of the lazy-greedy selection strategy described in the main text.

## 2.4 ENTROPY-DRIVEN SELECTION ALGORITHM (EDS)

The EDS algorithm selects a subset $S$ of size $|S| = s$ that maximizes KCE, addressing a combinatorial optimization problem. Below, we describe the optimization goal and the set-aware lazy-greedy algorithm used to achieve it efficiently, with theoretical justifications provided in Appendices B.5 and B.4.

### 2.4.1 OPTIMIZATION OBJECTIVE

The goal is to identify a subset $S$ that maximizes KCE:

$$S^* = \underset{S \subseteq \{1,\ldots,n\},|S|=s}{\arg\max} H(S).$$

This problem is computationally intractable due to its combinatorial nature, necessitating approximate strategies. We employ a submodular optimization approach, leveraging the diminishing returns property of KCE (see Appendix B.3).

### 2.4.2 SET-AWARE LAZY-GREEDY SELECTION

To maximize KCE efficiently, we define a concave-over-coverage objective:

$$F(S) = \sum_{j=1}^{m} w_j f(c_j(S)), \quad c_j(S) = \sum_{a \in S} \mathbf{B}_{a,j},$$

where $w_j \in \mathbb{R}_+^m$ are weights reflecting the importance of knowledge point $j$ (estimated from the dataset distribution), and $f$ is a concave, nondecreasing function. This objective is nonnegative, monotone, and submodular, ensuring that a greedy algorithm achieves a $(1 - 1/e)$ approximation to the optimal solution, as detailed in Appendix B.3. The subset selection is performed using the lazy-greedy algorithm (Algorithm 1). The choice of $f$ balances fidelity to KCE (using the entropy-derived $h$) and computational efficiency (using $\log(1 + x)$). Each marginal gain evaluation has complexity $O(\mathrm{nnz}(\mathbf{B}_{a,\cdot}))$, and the lazy-greedy approach scales efficiently with sparse matrices. An optional early stopping criterion, based on a revenue boundary, is discussed in Appendix B.4.

---

**Algorithm 1:** Lazy-Greedy EDS (Set-Aware Selection)

---

**Input:** Binary matrix $\mathbf{B} \in {0,1}^{n \times m}$; weights $w \in \mathbb{R}+^m$; budget $s$; concave $f$; tolerance $\varepsilon \geq 0$
**Output:** Selected indices $S$
$S \leftarrow \emptyset; c \leftarrow \mathbf{0}m$ ;                                   // Coverage counts
**for** $a \in 1, \ldots, n$ **do**
  Compute initial upper bound $U_a$ on $\Delta F(\emptyset; a)$; Push $(a, U_a)$ into max-heap $\mathcal{H}$;
**while** $|S| < s$ **do**
  $(a, U_a) \leftarrow \mathrm{PopMax}(\mathcal{H})$; // Exact marginal gain using current $c$
  $g_a \leftarrow \sum j : \mathbf{B}a, j = 1 w_j \big[ f(c_j + 1) - f(c_j) \big]; U_{\max} \leftarrow \mathrm{CurrentMaxKey}(\mathcal{H})$ (or $-\infty$ if
   empty); **if** $g_a \geq U_{\max} - \varepsilon$ **then**
    $S \leftarrow S \cup a$; **for** $j$ s.t. $\mathbf{B}_{a,j} = 1$ **do**
     $c_j \leftarrow c_j + 1$
  **else**
    $\mathrm{Push}(a, g_a)$ back into $\mathcal{H}$;
**return** $S$

---

### 2.4.3 WEIGHTED ENTROPY SCORING

To incorporate domain-specific priorities, we encode concept priorities with a weight vector $\mathbf{k} \in \mathbb{R}^m$ (e.g., from concept frequencies). For sample $a$, define $P_a = \frac{1}{m} \sum_{j=1}^{m} \mathbf{B}_{a,j}$ and $H(a) = -P_a \log_2 P_a$. The weighted score is

$$\mathrm{Score}(a) = H(a) \left( 1 + \gamma \sum_{i=1}^{m} k_i \mathbf{B}_{a,i} \right),$$

where $\gamma$ trades off diversity and importance. This heuristic steers greedy selection toward diverse samples emphasizing high-priority concepts. Estimation of $\mathbf{k}$ and single-pass variants are in Appendix 2.3.

## 3 EXPERIMENTS AND EVALUATIONS

We evaluate our entropy-driven data selection framework in two paradigms: supervised fine-tuning (SFT) for mathematical chain-of-thought (CoT) and retrieval-augmented generation (RAG). Baselines include QuRating (Wettig et al., 2024), SuperFiltering (Li et al., 2024a), Structure Entropy (Xie et al., 2024), random sampling, and the human-curated S1 subset (Muennighoff et al., 2025a). For RAG, we construct proprietary diabetes and general medical corpora and compare matched-size selections across methods. Ablations vary retrieval depth, corpus size, and top-$k$.

### 3.1 SUPERVISED FINE-TUNING EVALUATION BENCHMARK

We perform CoT SFT on the S1 data-ablation-full59K pool (Muennighoff et al., 2025b), using the human-curated 1k subset (simplescaling/s1K-tokenized) (Muennighoff et al., 2025b;a) as a high-

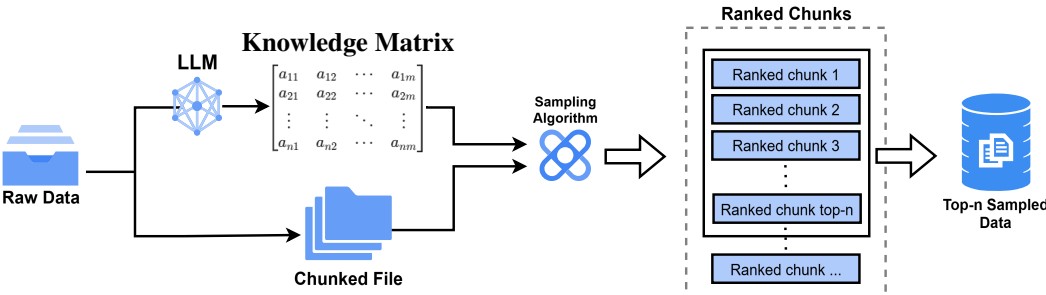

Figure 1: Overview of the Entropy-Driven Data Selection (EDS) workflow.

quality reference. Our method selects matched-size subsets (1k, 5k, 10k) under identical preprocessing and prompting, with baselines producing size- and token-matched counterparts. We fine-tune Qwen-32B-Instruct (Team, 2024) via standard next-token cross-entropy with consistent schedules across conditions. Selection is guided by knowledge coverage entropy (KCE), computed over a knowledge–sample matrix to balance per-sample uncertainty (row entropy) and global coverage, reducing redundancy and promoting diverse reasoning structures. The only difference across conditions is the upstream selection criterion.

We evaluate on the MATH-500 exam set using VLLM inference, reporting exact-answer accuracy. First, we compare KCE with non-negative knowledge-point weights to prioritize rare but critical units—against random sampling and the human-curated S1 subset. Next, we test alternative weighting schemes across baseline selectors by generating matched-size subsets and retraining under identical SFT protocols. We report overall accuracy, sample efficiency, and training stability.

## 3.2 CONSTRUCTION OF PROPRIETARY RAG CORPORA

We programmatically compile domain-relevant sources (diabetes: textbooks and clinical guidelines; general medical) (Holt & Flyvbjerg, 2024; Royal Government of Bhutan, Ministry of Health, Department of Medical Service, 2007; fun) and use LLMs to: (i) segment texts into atomic chunks, (ii) normalize to a controlled vocabulary of knowledge IDs, and (iii) finalize retrieval-ready passages with titles and structured metadata (knowledge IDs, source, language, timestamps). For each domain, we generate 1000 LLM-authored questions with automatic validation and light manual spot checks. We embed passages with BAAI/bge-large-zh and BAAI/bge-large-en (Chen et al., 2023; Xiao et al., 2023) and retrieve by cosine similarity (Salton et al., 1975) (top-$k$). Matched-size corpus variants are produced via our selection, QuRating, SuperFiltering, Structure Entropy, and the unselected full corpus.

To assess the selected corpora, we compute knowledge-point coverage rate Hit@k (the proportion of ground-truth knowledge points covered within the top-$k$ retrieved passages) and conventional MRR, and analyze the accuracy–efficiency trade-off as a function of corpus size. We first evaluate at top-10 retrieval, where each selection method operates at its theoretical data-efficiency point. We then vary (i) retrieval depth with $k \in \{5, 10, 20, 50\}$ and (ii) corpus size, always comparing under matched-size settings.

## 3.3 RAG EXPERIMENTS AND EVALUATION BENCHMARK

Let $Q$ be the query set with $|Q| = N$. For each query $q \in Q$, let $K(q)$ denote the required knowledge points and $R_k(q)$ the set of knowledge points covered by the top-$k$ retrieved entries (from annotated knowledge IDs).

The per-query knowledge-point hit rate at depth $k$ is:

$$\text{HitRate}_k(q) = \frac{|K(q) \cap R_k(q)|}{|K(q)|}. \tag{1}$$

The average knowledge-point hit rate is:

$$\text{AverageHitRate}_k = \frac{1}{N} \sum_{q \in Q} \text{HitRate}_k(q).$$

(2)

Define $r_k(q)$ as the smallest $r \in \{1, 2, \ldots, k\}$ such that the union of knowledge points covered by the top-$r$ retrieved entries contains all elements of $K(q)$. If no such $r$ exists within the top-$k$ entries, set $r_k(q) = 0$ by convention.

The per-query reciprocal rank is

$$\text{RR}(q) = \begin{cases} \frac{1}{r_k(q)}, & \text{if } r_k(q) \geq 1, \\ 0, & \text{if } r_k(q) = 0. \end{cases}$$

(3)

The average multi-point MRR (distinct from conventional MRR (Voorhees & Tice, 2000), as it requires setwise completion of $K(q)$) is:

$$\text{AvgMRR}_k = \frac{1}{N} \sum_{q \in Q} \text{RR}(q).$$

(4)

We compute knowledge-point coverage at depth $k$ (Hit@k) and the multi-point MRR, and also report conventional MRR for comparison. All configurations use identical embedding models, cosine-similarity retrieval, indexing, and query/annotation sets; selection methods differ only in the upstream criterion (KCE vs. baselines).

We then conduct two classes of experiments. (i) Fixed-size corpora: for each domain, we construct a matched-size evaluation corpus (Diabetes: 3K; Medical: 8K) for each selector and vary retrieval depth with $k \in \{5, 10, 20, 50\}$. (ii) Variable-size corpora: for each selector, we subsample 1%, 5%, 10%, 20%, and 50% of the full corpus and evaluate at multiple $k$. To operationalize the "revenue boundary," we sweep corpus-size–performance curves and select the smallest subset within 1% relative performance of the maximum Hit@10, yielding the data-efficiency point.

## 4 RESULTS

### 4.1 ENTROPY-DRIVEN SFT PERFORMANCE ON MATH-500

To evaluate our SFT data selection algorithm, we conducted experiments on the MATH-500 benchmark. Specifically, we compared 28 randomly sampled subsets with 28 entropy-selected subsets across different dataset sizes. All models were trained with full-parameter fine-tuning (see Table 7) and trained to convergence using an early stopping criterion (loss $\leq 0.05$ with a patience of 5). and inference was performed with the VLLM framework (Kwon et al., 2023), with the temperature fixed at 0 to eliminate stochastic variation. The model performance curves are shown in Fig. 2, and the complete performance results are summarized in Table 10. Across all dataset scales, entropy-based selection consistently outperforms random sampling, highlighting its ability to identify high-quality training data. Even relatively small entropy-selected subsets achieve performance comparable to much larger randomly sampled sets, demonstrating strong data efficiency. Notably, the entropy-selected subset reaches 450/500 at size 1000, closely matching the manually curated S1 dataset (452/500), and even exceeds it at size 500 (456/500). This consistent advantage across scales validates knowledge-point entropy as a principled and effective criterion for data selection.

The training loss trajectories are shown in Figure 4 for models trained on 40K and 50K samples, selected via entropy-based selection or random sampling. Entropy-selected subsets consistently converge faster and more stably, with the 40K subset exhibiting a steeper early decline, indicating stronger gradient signals from high-quality data. Notably, in this experiment, the 40K entropy-selected subset achieves slightly lower final loss than the 50K subset; this observation highlights the effectiveness of entropy-driven selection in identifying informative samples, rather than implying a general principle about optimal dataset size. Overall, entropy-driven selection delivers strong performance with fewer samples, demonstrating that principled data selection is an efficient and practical strategy for supervised fine-tuning (SFT) compared to indiscriminate dataset expansion.

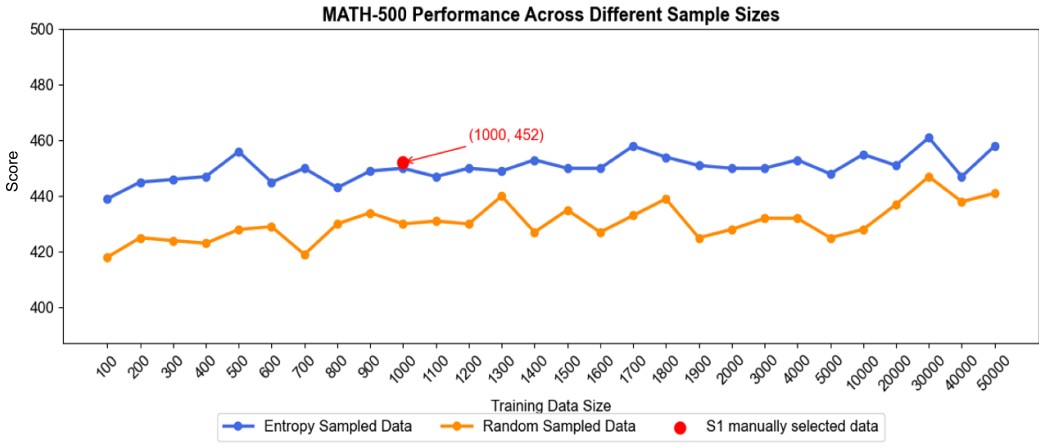

Figure 2: MATH-500 Performance with different trainind data size, The horizontal axis shows the dataset size, and the vertical axis shows the model's test scores. The red point at (1000, 452) indicates that the S1 team selected 1000 samples, achieving a score of 452 on the MATH-500 benchmark.

### 4.1.1 SFT EVALUATION OF BASELINE DATA SELECTION ALGORITHMS

We compared our method against several baseline algorithms under LoRA fine-tuning Table 7 across varying training data sizes, as shown in Figure 3. Overall, our KCE-based method consistently achieves higher exact answer accuracy than the baseline algorithms (Structure Entropy, QuRating, and SuperFiltering) at most dataset sizes, demonstrating its effectiveness in selecting high-quality, informative samples. Notably, KCE with knowledge-point weighting outperforms the unweighted variant in most cases (e.g., 455 vs. 444 at size 1000, 450 vs. 447 at size 2000), indicating that incorporating knowledge-point weights helps prioritize rare but critical knowledge units, further enhancing model performance. These results validate both the superiority of our entropy-driven selection method and the utility of weighted knowledge coverage for efficient and effective SFT.

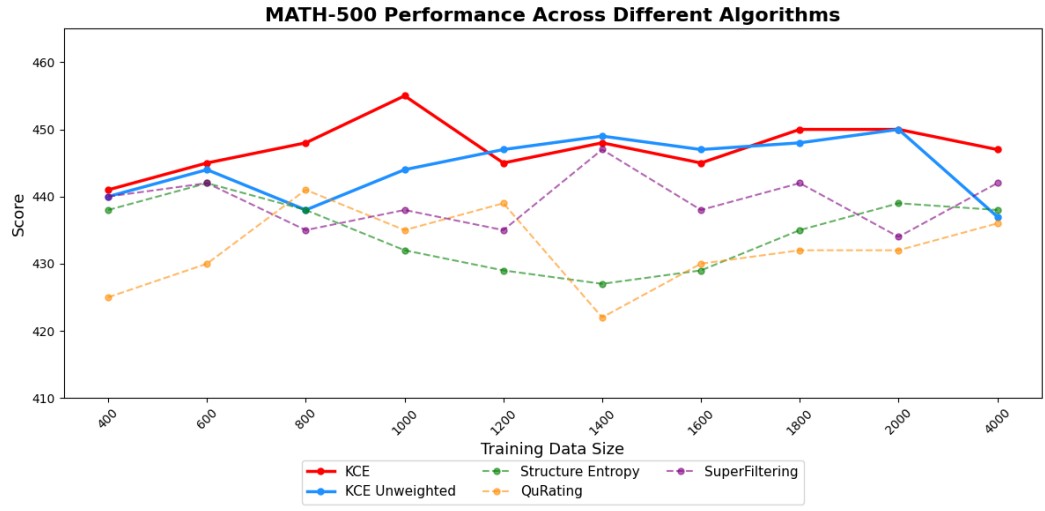

Figure 3: For LoRA fine-tuning, datasets of varying sizes were sampled using different algorithms. The red solid line represents KCE with knowledge-point weighting, the blue solid line represents KCE without weighting, and the remaining three dotted lines correspond to the other baseline data filtering algorithms.

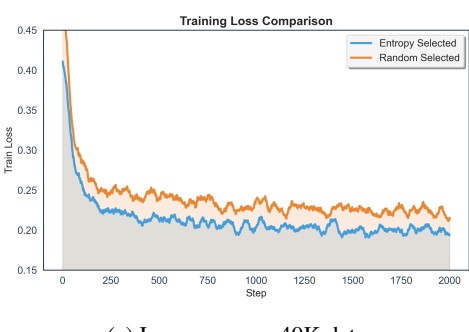 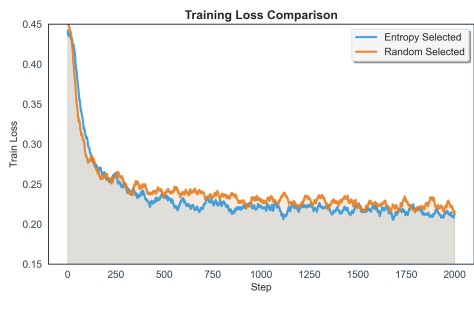

(a) Loss curve on 40K data                    (b) Loss curve on 50K data

Figure 4: Training loss curves for models trained with entropy-based selection (blue) and random selection (brown) on 40K (a) and 50K (b) datasets. Entropy-based selection accelerates convergence and achieves lower, more stable training loss compared to random selection.

## 4.2 RETRIEVAL EFFICIENCY ON MEDICAL KNOWLEDGE DATASETS

We evaluated our entropy-driven data selection framework, focusing on the proposed Knowledge-Centric Entropy (KCE) method, on two medical datasets: Diabetes and General Medical, comparing it with Qurating, Structural Entropy, and Superfiltering. The revenue boundary, illustrated in Figure 5, indicates where adding more samples yields diminishing returns, allowing redundant data to be discarded while preserving the most valuable knowledge. As summarized in Table 1, KCE consistently improves retrieval performance. On the General Medical dataset, which is high-dimensional with many sparse attributes and a less pronounced revenue boundary (as shown in Figure 5), KCE effectively prioritizes the most informative samples, leading to a notable improvement in MRR. Although the average coverage rate shows a slight decrease, it remains high, demonstrating that KCE enhances retrieval quality with minimal impact on overall coverage.

Table 1: Evaluation of data selection algorithms on RAG metrics. Reported metrics are average coverage rate and MRR. KCE achieves higher coverage and MRR with reduced dataset size compared to other methods.

| Dataset | Algorithm | Avg. MRR | Avg. Coverage Rate | Data Size |
|---------|-----------|----------|--------------------|-----------|
| Diabetes | Full Dataset | 0.4314 | 75.5% | 12K |
| | **KCE** | **0.4802** | **79.3%** | 3K |
| | Structure Entropy | 0.3372 | 61.9% | 3K |
| | QuRating | 0.3699 | 70.0% | 3K |
| | Superfiltering | 0.3695 | 68.7% | 3K |
| Medical | Full Dataset | 0.4511 | 73.9% | 20K |
| | **KCE** | **0.4685** | **72.9%** | 8K |
| | Structure Entropy | 0.3952 | 67.6% | 8K |
| | QuRating | 0.3992 | 68.7% | 8K |
| | Superfiltering | 0.4227 | 69.2% | 8K |

For the **Diabetes** dataset (251 Attributes), KCE achieves the highest coverage rate and MRR among all selection methods, increasing coverage from 75.5% to 79.3% and MRR from 0.431 to 0.480, while reducing the dataset size from 12K to 3K. In the **General Medical** dataset (1,122 Attributes), KCE maintains coverage, slightly decreasing from 73.9% to 72.9%, and further improves MRR from 0.451 to 0.468, despite a significant reduction in data size from 20K to 8K.

### 4.2.1 RETRIEVE WITH VARYING DATA SIZES

To evaluate the robustness of data selection algorithms under varying dataset sizes, we conducted experiments on the Diabetes and General Medical datasets using 1%, 5%, 10%, 20%, and 50% subsets. Overall, KCE demonstrates strong coverage and ranking quality across most scales. For instance, as shown in Table 2, on the Diabetes dataset, KCE attains 68.2%, 79.3%, 86.4%, and 90.1% coverage for the top 5, 10, 20, and 50 retrieved entries, generally outperforming Structure Entropy

and QuRating. While Superfiltering occasionally matches or slightly exceeds KCE at smaller subset sizes, KCE provides more consistent gains at larger scales. Similarly, on the Medical dataset, KCE achieves 62.6%, 72.9%, 81.6%, and 89.0% coverage for the corresponding top retrieved entries, highlighting its robustness in prioritizing high-value knowledge. These results indicate that KCE is effective and reliable even when dataset subsets are limited or sparse.

Table 2: RAG evaluation of different data selection algorithms across varying dataset sizes (% of full dataset). Metrics reported are average coverage rate (%) and MRR. Bolded entries indicate our proposed method (KCE) and do not necessarily correspond to the best-performing results.

| Dataset | Algorithm | Data Size (% of full dataset) | | | | |
|---------|-----------|------|------|------|------|------|
| | | 1% | 5% | 10% | 20% | 50% |
| Diabetes | **KCE** | **59.4** | **62.8** | **66.9** | **77.10** | **79.8** |
| | | **0.3620** | **0.3562** | **0.3679** | **0.4452** | **0.4885** |
| | Structure Entropy | 22.9 | 41.8 | 54.6 | 60.5 | 68.5 |
| | | 0.1385 | 0.2516 | 0.3102 | 0.3229 | 0.3959 |
| | QuRating | 17.1 | 33.0 | 50.2 | 62.1 | 71.3 |
| | | 0.08 | 0.17 | 0.2826 | 0.339 | 0.408 |
| | Superfiltering | 49.1 | 68.9 | 72.9 | 74.5 | 75.0 |
| | | 0.2443 | 0.3731 | 0.4083 | 0.4416 | 0.4342 |
| Medical | **KCE** | **32.5** | **48.2** | **67.0** | **73.5** | **74.8** |
| | | **0.1927** | **0.2825** | **0.3855** | **0.4568** | **0.4805** |
| | Structure Entropy | 22.1 | 46.7 | 58.0 | 63.7 | 69.5 |
| | | 0.1393 | 0.2828 | 0.3343 | 0.3835 | 0.4143 |
| | QuRating | 13.3 | 23.1 | 52.8 | 59.7 | 70.3 |
| | | 0.0850 | 0.1300 | 0.2973 | 0.3641 | 0.4301 |
| | Superfiltering | 41.4 | 50.9 | 65.9 | 70.0 | 73.9 |
| | | 0.2324 | 0.2816 | 0.3843 | 0.4288 | 0.4511 |

### 4.2.2 RETRIEVE WITH DIFFERENT TOP-K

In this experiment, we evaluated retrieval performance on fixed-size datasets (Diabetes top 3K and Medical top 8K) by varying the top-$k$ retrieved items from 5 to 50 to assess how well each algorithm ranks the most relevant knowledge. KCE consistently outperforms other methods across all top-$k$ settings. For example in Table 3, on the Diabetes dataset, KCE achieves Top@10 coverage of 79.3% with MRR 0.4802, compared to Structure Entropy (61.9% / 0.3372), QuRating (70.0% / 0.3699), and Superfiltering (68.7% / 0.3695). Similarly, on the Medical dataset, KCE attains superior coverage and ranking quality across Top@5 to Top@50 (e.g., Top@50 coverage 89.0% with MRR 0.4762), demonstrating its persistent advantage in prioritizing high-value knowledge over competing algorithms.

These results demonstrate that KCE consistently outperforms other algorithms in retaining essential knowledge and improving retrieval quality. By effectively prioritizing high-value information and removing redundancy, KCE enables substantial dataset reduction without sacrificing performance, reducing computational cost and enhancing retrieval-augmented generation on both low- and high-dimensional medical datasets.

### 4.3 REVENUE BOUNDARIES AND INFORMATION GAIN ACROSS DOMAINS

Entropy-based sampling improves data utilization efficiency on both Diabetes and General Medical datasets. Normalized entropy curves show that entropy-selected subsets achieve higher information gain per sample than unfiltered data, with a clear revenue boundary beyond which additional samples provide diminishing returns. In low-sample regimes, steeper slopes indicate faster acquisition of high-value data, while flattening slopes mark diminishing marginal returns and a natural stopping criterion. Across domains, entropy-based sampling consistently attains higher coverage efficiency than random selection, enabling the construction of compact, high-quality datasets for LLM training and retrieval-augmented generation.

Table 3: RAG evaluation of different data selection algorithms across varying top retrieval sizes (Top@5, 10, 20, 50). Metrics reported are average coverage rate (%) and MRR. KCE consistently achieves higher coverage and MRR than other methods across all top-k settings.

| Dataset | Algorithm | Retrieve Top Entries | | | |
| --- | --- | --- | --- | --- | --- |
| | | Top@5 | Top@10 | Top@20 | Top@50 |
| Diabetes top3K | **KCE** | **68.2** | **79.3** | **86.4** | **90.1** |
| | | **0.4655** | **0.4802** | **0.4853** | **0.4867** |
| | Structure Entropy | 50.1 | 61.9 | 74.9 | 85.6 |
| | | 0.3209 | 0.3372 | 0.3465 | 0.3510 |
| | QuRating | 54.4 | 70.0 | 81.6 | 89.5 |
| | | 0.3491 | 0.3699 | 0.3778 | 0.3808 |
| | Superfiltering | 54.6 | 68.7 | 78.2 | 89.5 |
| | | 0.3503 | 0.3695 | 0.3761 | 0.3798 |
| Medical top8K | **KCE** | **62.6** | **72.9** | **81.6** | **89.0** |
| | | **0.4539** | **0.4685** | **0.4737** | **0.4762** |
| | Structure Entropy | 57.0 | 67.6 | 76.0 | 84.9 |
| | | 0.3810 | 0.3952 | 0.4018 | 0.4044 |
| | QuRating | 57.4 | 68.7 | 77.4 | 85.4 |
| | | 0.3838 | 0.3992 | 0.4052 | 0.4079 |
| | Superfiltering | 58.0 | 69.2 | 77.1 | 84.7 |
| | | 0.4076 | 0.4227 | 0.4281 | 0.4308 |

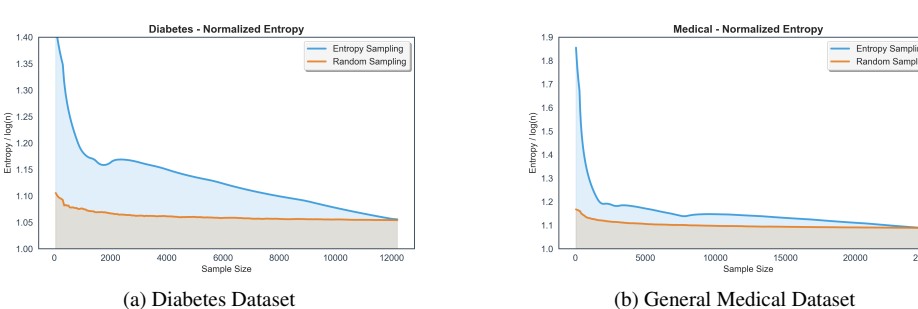

(a) Diabetes Dataset        (b) General Medical Dataset

Figure 5: Comparison of normalized entropy vs. sample size for two datasets using entropy sampling (blue line with orange variability) and random sampling (brown line). (a) Diabetes Dataset (0–12,000 samples, entropy 1.0–1.4). (b) General Medical Dataset (0–25,000 samples, entropy 1.0–1.9). Entropy sampling consistently yields higher normalized entropy than random sampling.

## 5 DISCUSSION

In this work, we propose a knowledge-centric data selection framework for domain adaptation, formalized through Knowledge Coverage Entropy (KCE) and instantiated via an entropy-driven, submodular selection algorithm (EDS). The approach models discrete knowledge units and prioritizes coverage diversity under cardinality constraints, aiming to reduce redundancy and improve sample efficiency in both supervised fine-tuning and retrieval-augmented generation in domain adaptation of large language models. Empirical results on MATH-500 and medical RAG indicate consistent gains with smaller datasets and more stable training dynamics.

## 6 REPRODUCIBILITY STATEMENT

To ensure reproducibility of our results, all code used for data processing, model training, and evaluation will be provided in a zip file as part of the supplementary materials. Detailed descriptions of the datasets, preprocessing steps, and experimental settings are included in the main text and appendices. This will allow readers to reproduce the reported experiments and verify the findings.

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

## A    USE OF LLM

The text of this article has been refined with the assistance of a large language model (LLM). All scholarly opinions, factual content, and final expressions remain the responsibility of the authors; the model was used solely to enhance the clarity, readability, and linguistic quality of the manuscript.

## B    LIMITATION

While the proposed entropy-driven Selection framework demonstrates promising results, it is not without limitations. First, the method assumes a certain degree of redundancy in the corpus, as the entropy computation relies on overlapping knowledge points across samples to establish informative distributions. Consequently, the approach may underperform on highly sparse datasets with minimal overlap. Second, the framework presumes that each information unit contains multiple knowledge points, providing sufficient variability to compute knowledge-point entropy. In cases where samples are extremely atomic—e.g., containing only a single knowledge point—the resulting knowledge matrix becomes nearly diagonal, rendering Knowledge Coverage Entropy computation ineffective.

Additionally, we currently represent the Information–Knowledge Matrix as binary, indicating whether a sample fully covers a knowledge point or not. While this simplification facilitates computation and aligns with the current entropy formulation, it neglects partial or graded coverage. We acknowledge this limitation and note that a probabilistic or weighted representation could better capture the degree of knowledge coverage in future work.

### B.1    ABLATION STUDY ON THE KNOWLEDGE MATRIX CONSTRUCTION

The ablation study of the knowledge matrix comprises two main parts. The first part examines the system's robustness to different frequency thresholds, defined as the minimum number of occurrences required for a knowledge point to be included, and evaluates the effect of weighted entropy on performance. The second part investigates the impact of using different scoring sources—LLMs versus human experts—on the construction of the knowledge matrix and analyzes whether these variations affect the algorithm's overall performance.

For this study, we use the GSM8K dataset (Cobbe et al., 2021) and train two model sets: DeepSeek-Distill-Qwen-7B and Qwen3-8B with 2k selected data (full set 7.9k) and LoRA adaptation (see Table 8). To evaluate the robustness of our data selection method, we compare performance across (1) different numbers of selected concepts k, (2) weighted vs. unweighted variants, and (3) multiple model scales. The results show that our method is highly stable across all dimensions. The detailed performances are list in Table 4 and Table 5

Table 4: Performance of different models under the weighted scoring scheme for six configurations of the knowledge frequency threshold $k$ (minimum occurrences of a knowledge point). The number in parentheses indicates the resulting number of knowledge units after applying the KCE module.

| Model | k=10(189) | k=15(123) | k=20(97) | k=25(78) | k=30(64) | k=50(45) |
|---|---|---|---|---|---|---|
| ds-distill-qwen-7B | 829 | 841 | 826 | 832 | 844 | 828 |
| qwen2-0.5B | 314 | 314 | 294 | 326 | 312 | 315 |
| qwen3-8B | 1206 | 1196 | 1192 | 1193 | 1206 | 1200 |

Table 5: Unweighted

| Model | k=10(189) | k=15(123) | k=20(97) | k=25(78) | k=30(64) | k=50(45) |
|---|---|---|---|---|---|---|
| ds-distill-qwen-7B | 810 | 789 | 808 | 787 | 808 | 803 |
| qwen2-0.5B | 320 | 334 | 316 | 310 | 317 | 315 |
| qwen3-8B | 1196 | 1207 | 1197 | 1203 | 1185 | 1203 |

First, performance remains nearly unchanged as k varies from 10 to 50, even though the number of selected concepts is reduced by more than 70% (from 189 to 45). The fluctuation in model accuracy stays within 1–5% across all models, demonstrating that our selection procedure is not sensitive to the exact choice of k.

Second, the weighted variant consistently exhibits lower variance than the unweighted version. This confirms that the weighting mechanism effectively filters out noisy or low-importance concepts, leading to more reliable improvements. For the smaller 0.5B model, the benefit of weighting is less pronounced, which is likely due to the limited capacity of small models to fully exploit the training data. Nonetheless, even in this low-capacity regime, the weighted (task-optimized) strategy remains more balanced acrSecond, the weighted variant consistently exhibits lower variance than the unweighted version.

Finally, the same trend holds across three models with very different capacities (0.5B, 7B, 8B), showing that the robustness of our approach is model-agnostic. The consistency across k, across weighting strategies, and across model scales provides strong evidence that our knowledge selection mechanism is inherently robust.

The ablation study on different knowledge extractors was conducted using the MedQA-CoT-LLaMA31 datasets (Jin et al., 2020; Gururajan et al., 2024). We constructed three versions of the knowledge matrix using Qwen-Max and Qwen-7B-Instruct and doctor rectified Qwen-max as the knowledge extractors. Since this dataset does not come with an established benchmark for evaluating the correctness of extracted knowledge points, we randomly sampled five subsets from the full training corpus as pseudo–test sets, each containing 1000 questions. The extraction accuracy on these five subsets is reported in Table 6. This training extracted 4k data out of 10K full set, and used the same LoRA config in previous study (See Table 8)

Table 6: Extraction accuracy of Qwen-Max and Qwen-7B-Instruct evaluated on five randomly sampled subsets (1000 questions each).

| Model | set1 | set2 | set3 | set4 | set5 |
|---|---|---|---|---|---|
| Qwen-max | 126/1000 | 132/1000 | 119/1000 | 119/1000 | 120/1000 |
| qwen-7B-Inst | 126/1000 | 128/1000 | 125/1000 | 125/1000 | 117/1000 |
| Rectified Qwen-max | 126/1000 | 132/1000 | 120/1000 | 119/1000 | 120/1000 |

Across the five sampled subsets, three extractors exhibit highly consistent performance with only minor fluctuations, indicating that our knowledge extraction pipeline is robust to model scale. Even though knowledge extracted by a smaller model may not be as precise as that from a larger model, the robustness of our knowledge matrix ensures that downstream training remains stable, which is particularly valuable in scenarios with limited computational resources. While the pseudo–test sets offer an approximate evaluation, incorporating human experts (e.g., medical professionals) would provide a more reliable assessment and capture domain-specific nuances beyond model capability, further improving the accuracy and trustworthiness of the resulting knowledge matrix.

## B.2 KNOWLEDGE COVERAGE ENTROPY DEFINITION AND BOUNDS

The Knowledge Coverage Entropy (KCE) measures the diversity of knowledge coverage in a subset $S \subseteq \{1, \ldots, n\}$ of size $|S| = h$ from a dataset represented by a binary matrix $\mathbf{B} \in \{0,1\}^{n \times m}$, where $\mathbf{B}_{i,j} = 1$ if sample $i$ covers knowledge point $j$, and 0 otherwise. To ensure numerical stability, we apply additive smoothing:

$$\mathbf{B}' = \mathbf{B} + \frac{\alpha}{nm}, \qquad \alpha = 10^{-6},$$

and normalize to obtain a joint probability distribution:

$$P_{i,j} = \frac{\mathbf{B}'_{i,j}}{\sum_{i=1}^{n} \sum_{j=1}^{m} \mathbf{B}'_{i,j}}.$$

The KCE for subset $S$ is defined as

$$H(S) = -\sum_{j=1}^{m} p_j \log_2 p_j, \qquad p_j = \frac{1}{h} \sum_{a \in S} \mathbf{B}_{a,j}.$$

The maximum entropy occurs when $p_j = 1/m$, yielding $H(S) \leq \log_2 m$. For the joint distribution over $nm$ outcomes, the upper bound is

$$H(S) \leq \log_2(nm) = \log_2 n + \log_2 m.$$

The normalized entropy is

$$H_n(S) = \frac{H(S)}{\log_2 h},$$

with $H_n(S) \leq 1 + \frac{\log_2 m}{\log_2 n}$. As $n \to \infty$, $H_n(S) \to 1$ (or less with redundancy). Redundancy in $\mathbf{B}$ (e.g., samples covering identical points) reduces $H(S)$ to $\approx \log_2 m$, exhibiting sublinear growth.

**Monte Carlo approximation.** To extend KCE to a potentially infinite-dimensional knowledge space $\mathcal{X}$, we replace the integral

$$H(S) = -\int_{\mathcal{X}} p(x) \log_2 p(x)\, d\mu(x), \qquad p(x) = \frac{1}{h} \sum_{a \in S} B_a(x),$$

with a Monte Carlo estimator. We draw $M$ samples $\{x_t\}_{t=1}^{M}$ from the base measure $\mu$, and approximate the entropy by

$$\hat{H}(S) = -\frac{1}{M} \sum_{t=1}^{M} \hat{p}(x_t)\, \log_2 \hat{p}(x_t), \qquad \hat{p}(x_t) = \frac{1}{h} \sum_{a \in S} B_a(x_t).$$

This estimator is unbiased and converges at a rate $O(M^{-1/2})$, independent of the dimensionality of $\mathcal{X}$.

**Knowledge-weighted variant.** When incorporating importance weights $k(x)$, the weighted entropy

$$H_k(S) = -\int_{\mathcal{X}} p(x) \log_2 p(x)\, k(x)\, d\mu(x)$$

is approximated by

$$\hat{H}_k(S) = -\frac{1}{M} \sum_{t=1}^{M} k(x_t)\, \hat{p}(x_t)\, \log_2 \hat{p}(x_t).$$

### B.3 SUBMODULARITY OF KNOWLEDGE COVERAGE ENTROPY

The effectiveness of the greedy algorithm relies on the submodular properties of KCE. Let $\mathbf{B} \in \{0,1\}^{n \times m}$ be the binary matrix, and $H(S) = -\sum_{j=1}^{m} p_j \log_2 p_j$ the KCE for subset $S$, where $p_j = \frac{1}{|S|} \sum_{a \in S} \mathbf{B}_{a,j}$. Although KCE is not strictly submodular, it exhibits diminishing marginal gains. For nested subsets $S_A \subseteq S_B$ and a sample $a \notin S_B$, the marginal gain satisfies

$$\Delta H(S_A; a) = H(S_A \cup \{a\}) - H(S_A) \ \geq\ \Delta H(S_B; a).$$

To derive this, consider the entropy function $H(p) = -\sum_j p_j \log_2 p_j$, which is concave in the probability vector $p$. When adding sample $a$ to subset $S$, define the coverage distribution induced by $a$ as

$$\delta_j = \frac{\mathbf{B}_{a,j}}{\sum_j \mathbf{B}_{a,j}}, \qquad c_a = \sum_j \mathbf{B}_{a,j},$$

and let $K(S) = \sum_{a \in S} \sum_j \mathbf{B}_{a,j}$ be the total coverage of $S$. The mixing parameter is

$$\lambda = \frac{c_a}{K(S) + c_a}.$$

The updated probability vector $p'$ is a convex combination:

$$p'_j = (1 - \lambda)p_j + \lambda\, \delta_j.$$

Since $H(p)$ is concave, by Jensen's inequality applied to the convex combination,

$$H(p') \ \geq\ (1 - \lambda)H(p) + \lambda H(\delta),$$

which yields the marginal gain bound

$$\Delta H = H(p') - H(p) \geq \lambda\big(H(\delta) - H(p)\big).$$

Because the Hessian of $H(p)$ is negative semi-definite, entropy changes are smaller when $p$ is near uniform (as in larger sets). For $|S_B| > |S_A|$, $K(S_B) > K(S_A)$, so $\lambda_B < \lambda_A$, and the distribution $p_B$ is closer to uniform, reducing $\Delta H(S_B; a)$. Alternatively, one can approximate $\Delta H \approx -D_{\mathrm{KL}}(p' \parallel p)$, where $D_{\mathrm{KL}}$ decreases with set size due to smaller $\lambda$, reinforcing the inequality for non-redundant $a$. This property supports the greedy algorithm's effectiveness, as detailed in the main text.

## B.4  INFORMATION GAIN AND REVENUE BOUNDARY

The information gain (IG) monitors the marginal contribution of adding samples to a subset. For a binary matrix $\mathbf{B} \in \{0,1\}^{n \times m}$ and subset $S_t$ of size $t$, the normalized entropy is $H_n(S_t) = H(S_t)/\log_2 t$, where $H(S_t) = -\sum_{j=1}^m p_j \log_2 p_j$ and $p_j = \frac{1}{t} \sum_{a \in S_t} \mathbf{B}_{a,j}$. The discrete information gain is

$$IG(t) = H_n(S_t) - H_n(S_{t-1}).$$

Due to diminishing returns (see Appendix B.3), $IG(t)$ decays as $t$ increases. The revenue boundary is defined as

$$t^* = \min\{t \; : \; IG(t) < \delta\},$$

where $\delta > 0$ is a task-specific threshold. To derive the decay, note that entropy is subadditive: for a new sample $a$ with row entropy $H(a) = -P_a \log_2 P_a$, where $P_a = \frac{1}{m} \sum_{j=1}^m \mathbf{B}_{a,j}$,

$$H(S \cup \{a\}) \leq H(S) + H(a),$$

and $H(a) \leq \log_2 m$ for uniform coverage. The marginal gain is

$$\Delta H = H(S \cup \{a\}) - H(S) \leq H(a).$$

Accounting for redundancy,

$$\Delta H = H(a \mid S) = H(a) - I(a; S),$$

where $I(a; S)$ is the mutual information measuring overlap. For large $t$, the expected $\Delta H_t \frac{\log_2 m}{t}$, as new samples cover at most $m/t$ new points on average (pigeonhole principle). Entropy concavity implies successive gains diminish:

$$\Delta H_t \leq \frac{\Delta H_{t-1}}{1 + \epsilon}, \qquad \epsilon > 0,$$

in redundant regimes. Summing the series,

$$H(S_t) = H(S_1) + \sum_{k=2}^t \Delta H_k \; \leq \; H(S_1) + \sum_{k=2}^t O\left(\frac{1}{k}\right) \; = \; H(S_1) + O(\log t).$$

Thus, $H_n(S_t) = O(1)$, and

$$IG(t) \approx \frac{\Delta H_t}{\log_2 t} = O\left(\frac{1}{t \log t}\right),$$

which asymptotically simplifies to $O(1/t)$. This decay justifies the revenue boundary for efficient stopping.

## B.5  MUTUAL INFORMATION APPROXIMATION

Maximizing KCE approximates maximizing mutual information $I(R; C)$ between samples (rows $R$) and knowledge points (columns $C$). Let $\mathbf{B} \in \{0,1\}^{n \times m}$ be the binary matrix, and $S \subseteq \{1, \ldots, n\}$ a subset. Define $R$ as a uniform random variable over $S$ and $C$ as a knowledge point conditioned on coverage. The joint entropy is $H(R, C) = H(S)$, where $H(S) = -\sum_{j=1}^m p_j \log_2 p_j$, $p_j = \frac{1}{|S|} \sum_{a \in S} \mathbf{B}_{a,j}$. The mutual information is

$$I(R; C) = H(R) + H(C) - H(R, C) = \log_2 |S| + H(C) - H(S),$$

where $H(R) = \log_2 |S|$ (uniform over rows) and $H(C) = -\sum_{j=1}^m P(\cdot, j) \log_2 P(\cdot, j)$, with

$$P(\cdot, j) = \frac{1}{|S|} \sum_{a \in S} \mathbf{B}_{a,j} \quad \text{(column marginals)}.$$

Maximizing $I(R;C)$ requires maximizing $H(C)$ (broad coverage) while minimizing $H(S)$ (low redundancy). Per-sample row entropy $H(a) = -P_a \log_2 P_a$, where $P_a = \frac{1}{m} \sum_{j=1}^m \mathbf{B}_{a,j}$, peaks at $P_a \approx 0.5$, favoring balanced samples that diversify $C$ and reduce $H(C \mid R)$. Under row independence, $H(S) = H(R) + H(C)$, so $I(R;C) = 0$; selection induces correlations, increasing $I$. The score

$$\text{Score}(a) = H(a) \cdot \left(1 + \gamma \sum_{i=1}^m k_i \mathbf{B}_{a,i}\right)$$

prioritizes task-relevant balance, approximating greedy $I(R;C)$ maximization (similar to submodular set cover).

## B.6 DATA DISTRIBUTION EFFECTS IN SUPERVISED FINE-TUNING

In supervised fine-tuning (SFT), let $z$ denote the logits, $p_\theta$ the predicted probability via softmax, $q$ the target distribution, and $L$ the cross-entropy loss (Ouyang et al., 2022):

$$L(\theta) = -\sum_{i=1}^m q_i \log p_{\theta,i}, \quad p_{\theta,i} = \frac{e^{z_i}}{\sum_j e^{z_j}}. \tag{5}$$

The gradient with respect to logits is

$$\nabla_z L = p_\theta - q. \tag{6}$$

The Fisher information matrix with respect to logits (Fisher, 1922) is defined as

$$F_z(q) = \mathbb{E} Y \sim q \big[(\nabla_z L(Y))(\nabla_z L(Y))^\top\big], \tag{7}$$

where $Y$ is a one-hot random variable drawn from $q$. Expanding this gives

$$F_z(q) = (p\theta - q)(p_\theta - q)^\top + \text{Cov}(Y). \tag{8}$$

Near convergence, $p_\theta \approx q$, so the rank-one term vanishes, and we have

$$F_z(q) \approx \text{Cov}(Y) = \text{diag}(q) - qq^\top. \tag{9}$$

The expected squared gradient norm is

$$\mathbb{E}[||\nabla_z L||2^2] = \text{Tr}(F_z(q)) = 1 - \sum i = 1^m q_i^2, \tag{10}$$

which is maximized for uniform $q$ (high diversity) and minimized for skewed $q$ (low diversity).

From this perspective, selecting datasets with high Knowledge Coverage Entropy (KCE) promotes a more uniform empirical knowledge distribution $p_j(S)$, ensuring that minibatches sampled from $S$ maintain high average gradient norms. This leads to faster and more stable convergence during SFT by avoiding overly skewed label distributions that would produce weak learning signals. In other words, maximizing row entropy $H(q)$ through KCE naturally aligns the data distribution to enhance both gradient strength and training efficiency.

### B.6.1 EFFICIENCY IN MODEL TRAINING

To validate the Revenue Boundary Theory, we prepared two sets of sampled datasets: (1) 28 subsets randomly sampled from the original dataset, with sizes ranging from 100 to 50,000; and (2) 28 subsets selected using the Entropy-Driven Data Selection algorithm. We trained 56 models in total using these datasets and visualized their performance trends.

### B.6.2 NORMALIZED ENTROPY AND INFORMATION GAIN

We conducted experiments on mathematical dataset by applying the proposed Entropy-Driven Data Selection algorithm to generate subsets with sizes ranging from 100 to 30,000. For each subset, we computed the normalized entropy and visualized its variation trend as the sample size increased. Furthermore, we plotted the information gain efficiency curves for both datasets to illustrate the points of maximum efficiency.

### B.7 ALTERNATIVE JOINT ENTROPY FORMULATION

An alternative joint entropy formulation is

$$H_{\text{joint}}(S) = -\sum_{i \in S} \sum_{j=1}^{m} P_{i,j} \log_2 P_{i,j},$$

where

$$P_{i,j} = \frac{\mathbf{B}'_{i,j}}{\sum_{i=1}^{n} \sum_{j=1}^{m} \mathbf{B}'_{i,j}}, \qquad \mathbf{B}' = \mathbf{B} + \frac{\alpha}{nm}, \ \ \alpha = 10^{-6}.$$

This accounts for row and column dependencies but is computationally costly and sensitive to redundancy. The marginal KCE in the main text is more efficient for diversity-focused selection.

### B.8 STOCHASTIC-GREEDY VARIANT

A stochastic-greedy variant samples a subset $R$ of size $r \approx \frac{n}{s} \log \frac{1}{\varepsilon}$ at each iteration, selecting

$$a^{\star} = \arg \max_{a \in R} \Delta F(S; a).$$

This achieves a $(1 - 1/e - \varepsilon)$ guarantee with reduced computational cost.

### B.9 HYBRID OBJECTIVE

A hybrid objective combines coverage and similarity:

$$F_{\text{hybrid}}(S) = \lambda \sum_{j=1}^{m} w_j f(c_j(S)) + (1 - \lambda) \sum_{x=1}^{n} \max_{a \in S} \text{sim}(x, a),$$

where $f$ is concave, and the second term is a facility-location function over a similarity graph. Both terms are submodular, preserving the $(1 - 1/e)$ guarantee of the lazy-greedy algorithm.

### B.10 PARAMETER SENSITIVITY ANALYSIS

We conducted a sensitivity analysis of KCE with respect to the smoothing parameter $\alpha$ and the weight balance $\gamma$ on sample sizes 500 and 1000. The normalized KCE ($H_n$) remains nearly constant across $\alpha \in [0.1, 2.0]$ and $\gamma \in [0, 1]$. For instance, with sample size 500, $H_n$ varies only from 1.2300 to 1.2320 ($< 0.2\%$), and with sample size 1000, from 1.2037 to 1.2062 ($< 0.3\%$). These small variations indicate that KCE is robust to both $\alpha$ and $\gamma$, and the algorithm reliably selects diverse knowledge subsets without significant sensitivity to hyperparameter choices.

### B.11 EMPIRICAL VALIDATION VIA SIMULATIONS

To empirically validate the decay in information gain, experiments were conducted on four datasets, with knowledge points $m$ ranging from 200 to 1000 and sample sizes $n$ between 20,000 and 60,000, averaged over 5 runs. Across all datasets, the normalized entropy grows sublinearly:

$$H_n(S) = \frac{H(S)}{\log m}, \qquad H(S) = -\sum_{i=1}^{m} p_i \log p_i,$$

where $p_i$ denotes the empirical frequency of knowledge point $i$ in the subset $S$. The information gain (IG) at step $t$ is defined as the marginal increase in entropy:

$$IG(t) = H_n(S_t) - H_n(S_{t-1}), \qquad S_t = S_{t-1} \cup \{x_t\}.$$

Empirically, $IG(t)$ starts high (approximately 0.99 at $t = 1$) and decays to near-zero (around $10^{-7}$ by $t = 1000$), following an overall $O(1/t)$ trend:

$$IG(t) \approx \frac{c}{t}, \qquad c > 0.$$

Moreover, we examine the slope of $IG(t)$, i.e., its discrete derivative:

$$\Delta IG(t) = IG(t+1) - IG(t).$$

On the diabetes dataset, the slope decreases from approximately

$$\Delta IG \approx -0.1 \times 10^{-4} \quad \text{(at the best advantage point)}$$

to

$$\Delta IG \approx -1 \times 10^{-6}, \quad \text{after which it stabilizes.}$$

This behavior confirms diminishing returns and validates the revenue boundary condition, where

$$IG(t) < \delta \quad \text{for } t \geq T_\delta.$$

Simulations on random binary matrices (e.g., $m = 251$, varying $n$) show $H_n(S)$ peaking early and $IG(n)$ decaying from $\sim 1.22$ to near-zero, confirming theorems. For entropy-selected subsets, $I(R; C)$ is 10–20% higher than random, tying theory to empirical wins.

### B.12 COMPUTATIONAL COMPLEXITY ANALYSIS

The lazy-greedy algorithm has time complexity that scales with the sparsity of the matrix $\mathbf{B}$. Each exact marginal gain evaluation is $O\big(\mathrm{nnz}(\mathbf{B}_{a,\cdot})\big)$, where $\mathrm{nnz}$ denotes the number of non-zero entries in row $a$. The lazy variant reduces the number of full evaluations by using upper bounds in the heap, leading to near-linear time in the total number of non-zero entries in $\mathbf{B}$ for sparse matrices. For dense matrices, the complexity is $O(nm \log n)$ in the worst case, but practical datasets are often sparse. The single-pass approximation is $O(nm)$, linear in the matrix size. Memory requirements are $O(n + m)$ for the heap and counts, making it scalable for large $n$ and $m$.

**Remarks and interpretation.**

- The quantity $1 - \sum_i q_i^2$ is closely related to the Gini impurity and measures the distributional uncertainty: it is zero for a one-hot (deterministic) $q$ and maximized when $q$ is uniform.

- The derivation above is performed in the *logit space*. For the Fisher information with respect to model parameters $\theta$, one needs to apply the Jacobian chain rule $F_\theta = J_{z \to \theta}^\top F_z J_{z \to \theta}$; nevertheless, the qualitative conclusion—uncertainty in $q$ increases the expected gradient magnitude—remains valid.

- The approximation $F_z(q) \approx \mathrm{Cov}(Y)$ relies on $p \approx q$. When the model is far from well calibrated, the additional term $(p-q)(p-q)^\top$ may be non-negligible and should be accounted for.

## C KNOWLEDGE DISTIL PROMPT

**Prompt Example:**

"You are a medical knowledge summarization assistant. "

"Task: Summarize the given text and extract only concise knowledge points directly related to diabetes. "

"Requirements:"

"1. Focus only on diabetes and its directly related aspects (symptoms, complications, treatments, risk factors, diagnostic methods, pathophysiology)."

"2. If there is **no diabetes-related content**, output exactly: NO"

"3. The output must consist only of short words or phrases (concise terms). "

"4. Do not output personal names, study names, or any content unrelated to diabetes. "

"5. Do not add external knowledge, only use the given content. "

"6. Output multiple knowledge points separated by commas, without extra text or explanations. "

"Example: In diabetes management, $\alpha$-glucosidase inhibitors may cause gastrointestinal side effects such as flatulence, abdominal discomfort, and diarrhea, particularly with high doses relative to carbohydrate intake, but these improve with gradual titration. Hypoglycemia is rare, and drug interactions are minimal, though concomitant use with motility agents or cholestyramine is not recommended. In the STOP-NIDDM trial, 31% of acarbose-treated patients discontinued early due to adverse effects compared to 19% with placebo."

"Response: $\alpha$-glucosidase inhibitors, gastrointestinal side effects, hypoglycemia rare, acarbose"

# D  KNOWLEDGE POINT EXAMPLE

| | | | |
|---|---|---|---|
| diabetes | cardiovascular disease | genetic factors | insulin resistance |
| insulin secretion | diabetes complications | diabetes management | diagnostic criteria |
| type 2 diabetes | GLP-1 | cardiovascular benefits | complications |
| diabetes care | HbA1c | type 1 diabetes | insulin |
| diabetic complications | cardiovascular risk | chronic kidney disease | continuous glucose monitoring |
| coronary heart disease | congestive heart failure | diabetic ketoacidosis | diabetic retinopathy |
| GIP | impaired fasting glucose | impaired glucose tolerance | glucose metabolism |
| hyperglycemia | beta-cell dysfunction | liraglutide | oral glucose tolerance test |
| endothelial dysfunction | oxidative stress | proliferative diabetic retinopathy | glucose monitoring |
| inflammation | neuropathy | retinopathy | type 1 diabetes mellitus |
| type 2 diabetes mellitus | thiazolidinediones | insulin sensitivity | urinary albumin excretion |
| Diabetes | polyuria | diabetic neuropathy | albuminuria |
| sulfonylureas | metformin | blood pressure management | microaneurysms |
| peripheral neuropathy | nephropathy | nausea | insulin deficiency |
| mortality | hepatic glucose production | blood glucose regulation | insulin production |
| glucose regulation | Diabetic retinopathy | diabetes treatment | insulin therapy |
| hypoglycemia risk | renal failure | diabetic nephropathy | Type 1 diabetes |
| MODY | pancreatic beta cells | hypertension | glucagon |
| lifestyle modifications | oral hypoglycemic agents | proteinuria | glycemic control |
| insulin pumps | rosiglitazone | pioglitazone | severe hypoglycemia |
| insulin use | cardiovascular risk reduction | microalbuminuria | blood pressure control |
| UKPDS | blood glucose control | insulin treatment | patient education |
| pregnancy | gestational diabetes mellitus | fasting plasma glucose | chronic hyperglycemia |
| microvascular complications | foot ulcers | macrovascular disease | weight loss |
| ketoacidosis | gestational diabetes | Type 2 diabetes | elevated blood glucose |
| increased diabetes risk | T1DM | T2DM | Type 1 diabetes mellitus |
| Type 2 diabetes mellitus | disease progression | obesity | physical inactivity |
| glucocorticoids | infections | Gestational diabetes mellitus | macrosomia |
| type 2 diabetes mellitus (T2DM) | fasting glucose | DCCT | islet autoantibodies |
| children | environmental factors | increased risk | islet autoimmunity |
| prevention | weight gain | Diabetic ketoacidosis | hypoglycemia |
| cardiovascular mortality | risk factors | macrovascular complications | vascular complications |
| metabolic syndrome | dyslipidemia | diabetes risk | physical activity |
| family history | $\beta$-cell dysfunction | free fatty acids | impaired insulin secretion |
| type 2 diabetes risk | diabetes prevalence | Diabetes prevalence | undiagnosed diabetes |
| smoking | age | glucose intolerance | coronary artery disease |
| early detection | adolescents | risk factor | stroke |
| infection | cardiovascular risk factors | end-stage renal disease | myocardial infarction |
| diet | quality of life | lifestyle interventions | weight reduction |
| ACE inhibitors | diabetes prevention | genetic predisposition | exercise |
| smoking cessation | Metformin | low- and middle-income countries | overweight |
| alcohol consumption | hyperinsulinemia | insulin administration | depression |
| combination therapy | blood pressure | comorbidities | glucose homeostasis |
| insulin release | hypoglycemia prevention | blood glucose levels | Sulfonylureas |
| $\beta$-cell function | DPP-4 inhibitors | exenatide | glucose uptake |
| lipolysis | insulin signaling | impaired glucose metabolism | skeletal muscle |
| TNF-$\alpha$ | adipose tissue | growth hormone | liver |
| gluconeogenesis | counter-regulatory hormones | monogenic diabetes | insulin secretagogues |
| NAFLD | atherosclerosis | neonatal diabetes | increased mortality |
| HbA1c levels | metabolic control | sulfonylurea | glucagon suppression |
| delayed gastric emptying | osmotic diuresis | vomiting | HbA1c reduction |
| glibenclamide | $\alpha$-glucosidase inhibitors | asymptomatic | cardiovascular disease risk |
| dehydration | heart failure | lifestyle changes | drug interactions |
| blood glucose monitoring | insulin dose adjustment | hypoglycemia unawareness | polydipsia |
| screening | HbA1c | self-management | diabetes education |
| autonomic neuropathy | gastroparesis | eating disorders | erectile dysfunction |
| diabetes self-management | blood glucose management | carbohydrate intake | insulin-treated diabetes |
| glycaemic control | cardiovascular events | cardiovascular outcomes | clinical trials |
| renal impairment | cognitive impairment | anxiety | Thiazolidinediones |
| GLP-1 receptor agonists | basal insulin | gastrointestinal side effects | depressive symptoms |
| older adults | urinary tract infections | islet transplantation | hyperglycaemia |
| hypoglycaemia | hypoglycaemia risk | SGLT-2 inhibitors | severe hypoglycaemia |
| glycaemic management | dyslipidaemia | semaglutide | |

# E  SUPPLEMENTARY TABLES

Table 7: Main Training Configurations and LoRA Parameter

| Training Configurations | | LoRA Parameter | |
|---|---|---|---|
| **Parameter** | **Value** | **Parameter** | **Value** |
| Learning Rate | $1 \times 10^{-5}$ | Learning Rate | $5 \times 10^{-5}$ |
| Epochs | 5 | Epochs | 5 |
| Method | Full Parameter | Method | LoRA |
| Model | Qwen2.5_32B_Instruct | PEFT | q,k,v,o,down,gate,up |
| Evaluation Dataset | MATH-500 | LoRA Rank | 64 |
| Early Stop Threshold | Loss $\leq 0.05$ | LoRA Alpha | 128 |
| Early Stop Patience | 5 steps | LoRA Dropout | 0.1 |
| Deepspeed Stage | Zero Stage 3 | Early Stop Threshold | Loss $\leq 0.15$ |
| | | Early Stop Patience | 5 steps |
| | | Deepspeed Stage | Zero Stage 3 |

Table 8: Ablation Training Configurations

| **Parameter** | **Value** |
|---|---|
| Learning Rate | $1 \times 10^{-5}$ |
| Epochs | 3 |
| Method | LoRA |
| PEFT | q,k,v,o,down,gate,up |
| LoRA Rank | 8 |
| LoRA Alpha | 32 |
| LoRA Dropout | 0.05 |
| Early Stop Threshold | Loss $\leq 0.2$ |
| Early Stop Patience | 3 steps |
| Deepspeed Stage | Zero Stage 1 |

Table 9: MATH-500 Performance Across Different Sample Sizes and different Algorithms on LORA fine tune

| Training Data Size | **KCE** | KCE Unweighted | Struct Entropy | QuRating | Superfiltering |
|---|---|---|---|---|---|
| 400 | **441** | 440 | 438 | 425 | 440 |
| 600 | **445** | 444 | 442 | 430 | 442 |
| 800 | **448** | 438 | 438 | 441 | 435 |
| 1000 | **455** | 444 | 432 | 435 | 438 |
| 1200 | **445** | 447 | 429 | 439 | 435 |
| 1400 | **448** | 449 | 427 | 422 | 447 |
| 1600 | **445** | 447 | 429 | 430 | 438 |
| 1800 | **450** | 448 | 435 | 432 | 442 |
| 2000 | **450** | 450 | 439 | 432 | 434 |
| 4000 | **447** | 437 | 438 | 436 | 442 |

Table 10: MATH-500 Performance Across Different Sample Sizes with full parameter fine tuning. The table reports scores for models trained on subsets selected by entropy sampling, random sampling, and the manually curated S1 dataset (1000 samples).

| Training Data Size | Entropy Sampled Data | Random Sampled Data | S1 Manually Selected |
|---|---|---|---|
| 100 | 439 | 418 | – |
| 200 | 445 | 425 | – |
| 300 | 446 | 424 | – |
| 400 | 447 | 423 | – |
| 500 | **456** | 428 | – |
| 600 | 445 | 429 | – |
| 700 | 450 | 419 | – |
| 800 | 443 | 430 | – |
| 900 | 449 | 434 | – |
| 1000 | 450 | 430 | **452** |
| 1100 | 447 | 431 | – |
| 1200 | 450 | 430 | – |
| 1300 | 449 | 440 | – |
| 1400 | 453 | 427 | – |
| 1500 | 450 | 435 | – |
| 1600 | 450 | 427 | – |
| 1700 | 458 | 433 | – |
| 1800 | 454 | 439 | – |
| 1900 | 451 | 425 | – |
| 2000 | 450 | 428 | – |
| 3000 | 450 | 432 | – |
| 4000 | 453 | 432 | – |
| 5000 | 448 | 425 | – |
| 10000 | 455 | 428 | – |
| 20000 | 451 | 437 | – |
| 30000 | 461 | 447 | – |
| 40000 | 447 | 438 | – |
| 50000 | 458 | 441 | – |

