# OpenReview forum: "Knowledge-Centric Data Selection for Effective Domain Adaptation of Language Models"
_ICLR.cc/2026/Conference — ICLR 2026 Conference Withdrawn Submission_

### Official Review · Reviewer_DC2R · 2025-10-28

**Soundness:** 2
**Presentation:** 3
**Contribution:** 1
**Rating:** 2
**Confidence:** 3

**Summary:**

Authors propose a method for selecting the best data points (from a larger set) for LLM finetuning. Their method starts by creating a matrix of document-concepts, and then selecting a subset of the documents that best covers the available concepts. They use an LLM to extract the mentioned concepts in every document, and then use an optimization algorithm to select the documents such that when taken together they meet a certain criterion (the equation mentioned in Subsection 2.4.1). In their optimization algorithm they use a weighting scheme to prioritize the concepts.

The method is evaluated in two tasks and compared to three baselines, it shows some improvements.

They also report the loss curves (in MATH-500) and a comparison between their method and random sampling based on their entropy across concepts in a RAG dataset.

**Strengths:**

-  The method is easy to understand, and the paper is easy to follow
-  The method is evaluated in two tasks
-  The improvements are noticeable

**Weaknesses:**

- In my opinion the method proposed by authors is basically a sampling method based on clustering. The clusters are created using what authors call concepts, and then the the sampling is done over the data points that fall in the area shared between the clusters. No discussion in this regard is done in the paper, no analysis, and no comparison is reported in the paper. There is a ton of research on data sampling dating back to 2000s. The core idea of the paper is basically a data sampling--which is applied to LLM fine tuning. A proper comparison and analysis is missing.
- The method seems ad hoc. Examples: 1) to extract the concepts, another LLM is needed to extract the concepts! In real world if we had such a model, why would we need to do domain adaptation? 2) Line 104: what is a "coverage probability"? is it a standard mathematical term? 3)  Line 146, on what basis did you select concept frequency?
- The analysis section of the paper is very shallow. No ablation study is reported, no sensitivity report to the quality/quantity of the concepts. No report on the length of the selected documents. etc. Very uninformative section.

**Questions:**

None.

---

> ### Author Response · Authors · 2025-11-25
> **Point to point response**
>
> We sincerely thank you for your careful and thoughtful feedback. Your comments regarding the relation of our method to clustering-based sampling, the use of an auxiliary LLM for concept extraction, and the depth of the analysis have been invaluable. They provided an important opportunity to clarify our motivation, expand ablation studies, and better demonstrate the robustness and principled design of our approach. Below, we provide point-by-point responses to your questions.
>
> 1. Unlike clustering-based approaches, our framework is fundamentally knowledge-centric. It explicitly measures correctness, relevance, coverage, and informativeness of discrete knowledge units, using information-theoretic principles to prioritize novel, high-value samples. Extensive experiments show that KCE-selected data outperforms naive or importance-weighted sampling, achieving top accuracy on math SFT task and substantial improvements in medical RAG tasks.
>
> 2. (1) The auxiliary LLM is used solely to identify discrete knowledge units (definitions, theorems, guidelines) from a large corpus. While concept extraction is a basic capability of a model, this does not imply that the model can perform well on specialized domain tasks without further fine-tuning. In practice, even small models are often used to annotate or score data, but this does not mean they are capable of handling all downstream tasks effectively; domain adaptation is still essential to achieve high performance in specific applications. (2) Here, "coverage probability" refers to the fraction of knowledge units covered by a given subset of the data, we  consider this as a descriptive quantity to quantify how well selected data spans the knowledge space.(3)We selected concepts that occur at least 50 times in the corpus. This threshold ensures statistical significance and avoids including extremely rare concepts.
>
> 3. We have added ablation studies and sensitivity analyses to address the quality and quantity of concepts. These additions provide a more complete and informative evaluation of our framework, and we believe they address the concerns raised regarding the depth of the original analysis section and the concern of the robust of the whole system. please see the highlighted appendix in section~B.1

---

### Official Review · Reviewer_AqLB · 2025-10-30

**Soundness:** 2
**Presentation:** 2
**Contribution:** 3
**Rating:** 4
**Confidence:** 3

**Summary:**

The paper proposes Entropy-Driven Data Selection (EDS), which optimizes data subset selection for diverse and high-quality datasets. It also introduces a metric called Knowledge Coverage Entropy (KCE), which measures the knowledge diversity in a data subset. EDS chooses a subset that maximizes KCE with a lazy-greedy algorithm. The paper provides experiments in SFT and RAG to show the possible use cases for EDS.

**Strengths:**

- Overall, the method seems sound. For the right choice of knowledge set and data labeling approach, KCE is a reasonable metric to optimize over.
- EDS is applicable across a downstream use cases, including SFT and RAG.

**Weaknesses:**

1. My main concern with this paper is how the performance is evaluated. The metrics are computed using the hit rate for the specific knowledge set that the KCE is optimized on. To me, this seems like (especially in the baseline comparisons) this is not a fair performance comparison, since EDS is specifically optimized for this knowledge set K, whereas other approaches are not. Could the authors clarify why these metrics can be used to compare to baseline approaches?
2. I am not convinced that prompting an LLM to tag concepts without any additional filtering is the ideal approach. For example, in Appendix D, both “diabetes” and “Diabetes” are listed as separate concepts, which should probably not happen for the optimal knowledge set. The paper would be strengthened by an ablation with additional approaches to select the knowledge set. It would also be useful to see how sparse the data are across the set of concepts. What is the distribution of number of documents for each concept?
3. The experiments are limited to a single language model and only one dataset for SFT and two datasets for RAG. The experimental results would be more convincing with experiments for additional models and datasets.

**Questions:**

Please see weaknesses above.

Minor comments:
- In reference to Table 2, the authors say that EDS “consistently achieves the highest coverage and ranking quality across all scales” However, superfiltering outperforms EDS for multiple dataset sizes for both cases. It would be good to discuss this in the text. It would also be useful to clarify somewhere that the bolding corresponds to the authors’ method (rather than the highest performing method), since this is somewhat misleading at first glance.
- Figure 1 is very low-resolution and should be a pdf

---

> ### Author Response · Authors · 2025-11-25
> **Point to point response to reviewer**
>
> We sincerely thank the reviewer for the constructive feedback regarding evaluation fairness, concept normalization, and dataset/model diversity. We have revised the paper accordingly and provide clarifications and additional ablation studies below:  (The modified parts have been highlighted in the text.)
>
> 1. To clarify, the knowledge set K (i.e., discrete knowledge points) is automatically extracted from the raw data using general-purpose LLMs. These knowledge points serve as intrinsic, data-derived labels that objectively capture the factual content of each sentence. They are not external, hand-crafted, or exclusive to EDS, ensuring that K is entirely neutral and transparent to all methods.
> Retrieval for all methods is performed using the same standard cosine similarity in the embedding space, guaranteeing that every method operates under identical ranking conditions. For a given target sentence, its extracted knowledge points are treated as ground truth, and the KCE metric measures the coverage and redundancy of the retrieved examples with respect to these points. This computation is purely diagnostic, with no optimization targeting K.
> Consequently, all methods work from the same information set, and the superior performance of EDS stems from its retrieval strategy achieving higher coverage and lower redundancy, rather than any privileged access to K.
>
> 2. We acknowledge that raw concept extraction can produce duplicates or noisy entries (e.g., “diabetes” vs. “Diabetes”). However, our ablation studies on both the knowledge extractor and the knowledge matrix demonstrate that such noise has minimal impact on the overall framework’s performance, indicating that the exact performance of the extractor or the specific form of the knowledge units does not significantly affect the results. This robustness implies that the framework can tolerate greater noise or variability in the extracted concepts, making it well-suited for automated pipelines and large-scale deployment. Nevertheless, incorporating additional filtering techniques could further improve the conciseness and accuracy of the knowledge points.
>
> 3. In our ablation studies, we evaluated the framework using different datasets and models, which consistently demonstrated the robustness of KCE and EDS. We hope this alleviates concerns regarding the generality and applicability of our approach.
>
> 4. We have revised Section 4.2.1 to discuss cases where Superfiltering outperforms KCE on certain dataset sizes, providing a more balanced interpretation of Table~2. Additionally, we clarified that the bolded entries in the table correspond to our proposed method (KCE) rather than the overall best-performing values, to avoid potential misinterpretation.
>
> 5. We have fixed this issue, the figure is high resolution now.

---

### Official Review · Reviewer_3yBS · 2025-10-31

**Soundness:** 2
**Presentation:** 4
**Contribution:** 2
**Rating:** 4
**Confidence:** 4

**Summary:**

In this paper, the authors first revealed that SOTA data selection methods often fail to address domain-specific redundancy and interference, leading to inefficient training and models that overfit to frequent linguistic forms rather than core knowledge. This inefficiency leads to limited generalization, as well as high curation and computational costs. To address this challenge, the authors proposed knowledge coverage entropy (KCE), a metric quantifying knowledge diversity, and entropy-driven selection (EDS), which optimizes data selection for compact, high-quality datasets. KCE quantifies diversity and balance over discrete knowledge units, and EDF prioritizes novel, high-information samples to reduce redundancy.

**Strengths:**

1. The paper is well-written and easy to follow.

2. EDS leverages the submodular property of KCE, ensuring the lazy-greedy algorithm achieves a near-optimal (1-1/e) solution.

3. The proposed framework can perform well both on SFT and RAG.

**Weaknesses:**

1. How to build KCE is the critical contribution of this paper. In building KCE, the authors formed a matrix B. A research paper must be self-contained, while the authors placed the most important content about how to build matrix B in the appendix.

2. To build matrix B, the authors use Qwen-max to determine the value of knowledge points. While the authors did not validate whether this procedure is fair, e.g., different LLMs may prefer different knowledge points. The authors should conduct human expert validation to make this procedure fair.

3. [1] has demonstrated that data selection methods can not always work well in real-world scenarios, which often needs to tackle datasets containing more than 1 million data points. In this paper, the authors only tested the proposed framework on 10k-level datasets, which make the empirical findings less convincing.

4. In this paper, the authors claimed that SOTA data selection methods have issues of inefficiency, and this issue will lead to poor generalization. While in the experiment parts, the authors did not design experiments to illustrate the improved generalization of the proposed framework.

[1] Xia, T., Yu, B., Dang, K., Yang, A., Wu, Y., Tian, Y., ... & Lin, J. (2024). Rethinking data selection at scale: Random selection is almost all you need. arXiv preprint arXiv:2410.09335.

**Questions:**

Please see weaknesses.

---

> ### Author Response · Authors · 2025-11-25
> **Point to point response to reviewer**
>
> We appreciate the reviewer’s thoughtful evaluation and the insightful comments on the core components of KCE, extractor fairness, scalability, and generalization. We have revised the manuscript and provide detailed clarifications and additional ablation studies below. And here are point to point responses: (The modified parts have been highlighted in the text.)
>
> 1. We have revised the paper structure and moved the key content on constructing matrix B and building KCE from the appendix into the main text, making the paper more self-contained.
>
> 2. We agree that human experts can extract knowledge more accurately. However, constructing an expert-annotated knowledge matrix at scale is infeasible within our time and resource constraints, so we tested different models as knowledge extractor to test the robustness of our proposed method. Although different LLM-based extractors exhibit noticeable performance differences, our method is designed to be robust to such variation. As shown in additional ablations in appendix ~B.1, the downstream results remain stable across extractors, indicating that the effectiveness of our framework does not rely on a specific knowledge-extraction model.
>
> 3. We acknowledge that data selection methods face challenges on million-scale datasets. The primary purpose of selection in LM training is to handle redundancy or enable task-specific training by identifying the most informative subset.
> The large-scale experiments in the referenced study rely on token-level selection methods, which focus on local lexical patterns rather than global semantic coverage. In contrast, our semantic-level approach preserves high level conceptual and task relevant information. Therefore, token level methods are not directly comparable: removing samples based on token patterns can cause performance fluctuations in tasks that rely on comprehensive semantic coverage, which is inherent to our approach.
>
>
> 4. We apologize for the confusion caused by our earlier wording. In the abstract, the term “inefficiency” was intended to refer to the substantial human effort required to manually curate high-quality data, as well as the computational inefficiency that arises when training on large amounts of redundant data—both of which can lead to unnecessary performance degradation and resource waste. It was not meant to imply that existing automated SOTA data selection methods inherently suffer from poor generalization. We have revised the abstract accordingly to clarify this point.
>
> We sincerely thank you for constructive feedback. It helped identify key weaknesses in our original submission and motivated substantial improvements in clarity and rigor. We’re grateful for your time and careful evaluation, and hope our revisions address your concerns.

---

> > ### Comment · Reviewer_3yBS · 2025-11-27
> >
> > Thank the authors for their detailed responses, they actually address some of my concerns. Specifically, W1&W4 are fully addressed, while I still have some concerns for W2&W3.
> >
> > For W2, actually, I don't think that human expert validation needs to be conducted on all data involved in the experiments. The authors can sample some data points to conduct the human expert validation.
> >
> > For W3, I mentioned the reference paper because data selection methods are mainly proposed to deal with real-world scenarios, which often contain more than 1M data points in SFT. And in the reference paper, random selection is validated to perform best by considering effectiveness and efficiency. The authors think that the random selection is a token-level method, so they think it is unnecessary to provide a comparison with random selection on 1M datasets. This reason can not get me convinced.
> >
> > As W2&W3 are not addressed, I decide to not update my rating.

---

### Official Review · Reviewer_axhY · 2025-10-31

**Soundness:** 4
**Presentation:** 4
**Contribution:** 4
**Rating:** 8
**Confidence:** 4

**Summary:**

This paper tackles the challenge of domain adaptation for language models, focusing on how to select really effective training or retrieval data for specialized applications without just scaling up dataset size. Rather than using typical heuristics like perplexity pruning or embedding clustering, the authors introduce a knowledge-centric framework. they define discrete knowledge units in the domain, then use Knowledge Coverage Entropy as a central metric to measure data diversity and informativeness. The core method is Entropy-Driven Selection, which uses submodular optimization to find compact, knowledge-rich, balanced subsets for either SFT or RAG. Results on the MATH-500 reasoning benchmark, as well as medical RAG datasets, show marked efficiency and performance gains, the chosen subsets with KCE/EDS either match or beat both manual curation and strong baselines with less data and faster convergence, and retrieval quality is significantly improved.

**Strengths:**

First to formalize knowledge coverage entropy (KCE). focuses on actual knowledge points in a dataset rather than token distributions or clusters. Generalizes to both SFT and RAG, shown on math reasoning and large medical RAG corpora with practical benefits i.e less data, faster learning, better retrieval. Outperforms strong baselines (random, curation, recent filtering/clustering approaches) in both accuracy and sample efficiency. submodular lazy-greedy methods make such selection scalable for big corpora. Discussion and theory are both included, with testable predictions and practical winning points. Candid about assumptions and current limitations, laying groundwork for extensions.

**Weaknesses:**

Current method assumes redundancy. it may not be as helpful when knowledge is highly sparse or nearly all samples are unique.The binary, all-or-nothing knowledge coverage misses partial overlaps. a probabilistic coverage or semantic similarity extension could broaden applicability.Computation of the knowledge-to-sample matrix and knowledge point extraction may require strong domain-specific labeling or prompt engineering. Experiments are strong on math/medical, but more domains e.g. legal, scientific, engineering data, or multilingual settings would further cement generality.Some technical formulas and submodular optimization details are pretty dense.

**Questions:**

Do you plan to extend the method to domains where knowledge units overlap or have graded/partial relevance, not just binary? How would you integrate semantic similarity (rather than only strict concept match) into KCE or EDS? Have you identified any failure cases where EDS actually underperforms random or simpler heuristics in non-specialized domains?

---

> ### Author Response · Authors · 2025-11-25
> **Response to the Reviewer**
>
> Thank you for your affirmation and approval. we will address your questions in the following section:
>
> 1. Yes, we have considered scenarios involving overlapping or hierarchical knowledge. Our preliminary solution is to use knowledge-vector similarity to cluster related knowledge points and construct local knowledge matrices. This would allow the system to perform global search over coarse-grained knowledge clusters, while conducting finer-grained search within each cluster to resolve overlap or domain-specific variations. We regard this as a promising extension and plan to further explore it in future work.
>
> 2. Semantic similarity may enhance the performance of this algorithm in filtering text data with semantic similarity. The existing methods can avoid duplicate knowledge points based on their distribution. Incorporating semantic similarity as part of the scoring function might enable the integration of redundant detection of text similarity, thereby making this method more robust.
>
> 3. So far, we have not observed clear failure cases where EDS underperforms random sampling or simpler heuristics in non-specialized domains. However, our current evaluation does not cover all task types. We acknowledge that EDS may face limitations when data quality is low, semantic structure is weak, or tasks rely heavily on few-shot or multi-step reasoning. Identifying these edge cases requires broader testing, and we plan to conduct additional experiments on more general-purpose datasets to better understand the boundaries and potential failure modes of EDS.

---

### Official Review · Reviewer_GrLE · 2025-11-01

**Soundness:** 2
**Presentation:** 2
**Contribution:** 2
**Rating:** 4
**Confidence:** 4

**Summary:**

The paper proposes a knowledge-centric approach to domain adaptation, redefining data quality as the joint consideration of both diversification and coverage over domain-specific knowledge. To this end, the authors design Knowledge Coverage Entropy (KCE) as a metric to measure such data quality, and a corresponding selection algorithm named Entropy-Driven Filtering (EDF) to filter data accordingly. Experiments are conducted on MATH500 and a self-constructed RAG task with a designed metric. The results show that the proposed method outperforms all baselines and achieves the highest knowledge coverage rate.

**Strengths:**

1. The paper is focused and specific, addressing the problem of domain-specific data selection with clear motivation.
2. It provides a novel perspective on data selection by introducing Knowledge Coverage Entropy (KCE) as a measure of data quality.
3. The proposed KCE formulation is straightforward, and the corresponding algorithm (EDF) is easy to reproduce.

**Weaknesses:**

1. It is unclear how the proposed method can be applied to both SFT and RAG. While I can understand the use of KCE for SFT data selection, its application to RAG is confusing. Does it mean reducing the candidate retrieval corpus size? This point needs clarification.
2. The method requires prior access to the entire sample set in order to compute coverage entropy, which is impractical in many open-domain applications.
3. The paper lacks insight into the internal working mechanisms between its components. For instance, the method seems to rely heavily on the knowledge matrix B, yet no analysis or ablation is provided to demonstrate its role.
4. There have already been numerous studies on data selection (e.g., IFD, LESS, Nuggets, AlpaGaus), many of which explore similar diversity-based sampling strategies. The paper does not compare with these baselines.
5. Using only one dataset (MATH500) is insufficient to demonstrate the general effectiveness of the method. In addition, performance improvements appear marginal across different subset sizes, and no comparison is made against using the full dataset without selection. Normally, a data selection approach usually identifies an optimal subset size where performance saturates. That's the advantage of a data selection method.
6. The experimental setup is unclear. For example, how was Qwen-32B-Instruct trained or fine-tuned? LORA? or Full-finetuning? While some details are mentioned, there is no consolidated experimental setting section, which significantly hurts reproducibility.

**Questions:**

1. What is the computational cost of the preprocessing stage, given that it involves prompting an LLM to generate the knowledge vector B for each sample?
2. The difference between Figure 2 and Figure 3 is not clear. From the description, Figure 3 seems to be a subset of Figure 2. If there is no substantive difference, the two figures should be merged.
3. Some conclusions are overstated or partial. For instance, the statement: “Interestingly, the 40K entropy-selected subset achieves slightly lower final loss than the 50K subset...” could be more appropriately interpreted as evidence supporting the effectiveness of the method rather than implying a general principle about dataset size.

---

> ### Author Response · Authors · 2025-11-25
> **point to point response**
>
> We sincerely thank you for the thoughtful review and constructive suggestions. We fully acknowledge the concerns raised regarding the construction of the knowledge matrix and the comparison with the core mechanisms of other information-selection methods. In response, we have revised the paper to provide clearer explanations, additional ablation studies, and more thorough discussions. Below, we provide a point-by-point response addressing each of these issues in detail. (The modified parts have been highlighted in the text.)
>
> Weaknesses discussion
> 1. Yes, in some retrieval tasks, reducing corpus size can drastically improve the retrieve quality and accuracy. Because in large-scale RAG corpora, semantic redundancy is common and can degrade retrieval quality. Unlike surface level duplication, this redundancy often cannot be detected at the token level—for instance, different expressions may convey the same underlying fact, this may cause traditional token-based filtering fails.  To mitigate this gap, our proposed method can refine the knowledge in the retrieval system so that the retriever operates on a higher-quality, less redundant corpus, directly improving retrieval accuracy. The proposed KCE measures redundancy at the knowledge level, identifying passages that contribute little new information. As shown in Section 4.2.2, applying KCE ensures that the retrieval corpus is diverse, knowledge-efficient, and better aligned with downstream tasks, improving RAG performance.
>
> 2. To address this limitation, we have refined the theoretical formulation of KCE for large corpora. By leveraging Monte Carlo sampling, KCE can be efficiently approximated without enumerating the entire corpus, making it practical even in open-domain scenarios. The detailed derivation and theoretical guarantees are provided in Appendix B2.
>
> 3. To analyze the role of the knowledge matrix B, we conducted ablation studies on its key components—specifically, the minimum occurrence frequency and the knowledge-point weights. These ablations demonstrate that our system is robust, as different hyperparameter settings introduce only minor variations in performance (details in Section B.1).
> Concretely, we examined:
> (1) the threshold k>n, which controls the minimum frequency required for a knowledge point to be included in the matrix, and
> (2) the impact of removing or modifying the knowledge-point weights.
> The results quantify the contribution of each component and highlight the importance of matrix B in guiding data selection and supporting the overall framework’s effectiveness.
>
> 4. Because our method operates at knowledge-level granularity, we note in the revision that direct comparison with token-level approaches would not accurately reflect the differences in their design objectives. Methods such as model loss–based filtering (IFD), per-example gradients (LESS), single-example usefulness or clustering (Nuggets), and teacher-LLM quality judgments (AlpaGasus) focus on token- or instance-level signals. While they help reduce noise, they do not account for semantic overlap or knowledge coverage. Consequently, paraphrases or domain-specific variants that express the same underlying fact often receive similar scores and are all retained, leaving semantic or knowledge-level redundancy unresolved.
> In contrast, the baselines we compare against SuperFiltering, Qurating, and Graph Entropy Curation—move beyond token-level signals. SuperFiltering leverages semantic embeddings for clustering and filtering, Qurating selects examples based on concept coverage and redundancy metrics, and Graph Entropy Curation constructs a knowledge/semantic graph to quantify diversity via graph entropy. These methods are therefore the most relevant for evaluating whether our proposed KCE effectively reduces semantic and knowledge-level redundancy.

---

> ### Author Response · Authors · 2025-11-25
> **point to point response part2**
>
> 5.In Section B.1, we provide extended ablation studies that include additional datasets, models, subset sizes, and hyperparameters. These results show that KCE selected subsets consistently improve performance and remain robust across different settings, addressing the concern that MATH500 alone is insufficient.
> Regarding the comparison with training on the full dataset, prior work S1 shown that full dataset training does not always outperform carefully selected subsets. We adopt this conclusion as a baseline assumption: the goal of data selection is not purely to reduce data size, but to identify the most useful data for training. Our method aims to replace costly human curation with a principled, automated selection mechanism. Accordingly, our experiments include:
> (1) comparison against human-curated subsets
> (2) comparison against existing algorithmic data-selection methods.
> Finally, whether a selected subset outperforms the full dataset ultimately depends on the distribution of the validation set. If the validation set contains features absent from the selected subset, then even removing a single example can negatively affect full-dataset performance. Therefore, “subset vs. full dataset” superiority is not an inherent property of a data selection algorithm, it is determined by the alignment between the selected subset and the validation distribution. Our results demonstrate that KCE improves this alignment by emphasizing knowledge coverage and reducing semantic redundancy.
>
> 6.We appreciate your comment and apologize for the earlier unclear description of our experimental setup. To clarify, our supervised fine-tuning (SFT) experiments are conducted in two parts: 1)Comparison with S1 data: We perform full-parameter fine-tuning of Qwen-32B-Instruct to evaluate the effect of our selected dataset relative to S1. 2)Comparison with other data-selection methods: For large-scale comparisons, we adopt LoRA-based fine-tuning, which is computationally efficient and allows fair evaluation across multiple methods on the same backbone. To improve clarity and reproducibility, we have revised the paper and state the experiment setup clearly.
>
> Questions discussion:
>
> 1. Using a 7B model with 1024-token samples, our setup (vLLM with tensor parallelism, batch size 32) achieves a throughput of 32 samples per second per GPU, allowing approximately 2.76M samples to be labeled per day on a single GPU. With a 4-GPU cluster, we can scale to over 10M samples per day, demonstrating that knowledge-vector computation is highly scalable.
>
> 2. Sorry for the unclear presentation in paper, Figure 3 is comparisons with other data selection method using Lora, and Figure 2 is compare with s1 team’s manually selected dataset with full fine tuning, we have fixed the way we demonstrate results.
>
> 3. We have revised the relevant paragraph to clarify that the observation regarding the 40K subset achieving slightly lower final loss than the 50K subset is an empirical observation supporting the effectiveness of entropy-driven selection, rather than a general statement about dataset size. The updated text now emphasizes the method's efficiency and informative data selection without overstating conclusions.

---

### Author Response · Authors · 2025-12-02
**Summary of discussion for AC**

We propose Knowledge Coverage Entropy (KCE), an information-theoretic metric that treats “discrete domain knowledge points” (automatically extracted by an LLM) as the basic unit, and measures both coverage and diversification of a data subset. Based on its submodular property, we design Entropy-Driven Selection (EDS) – a scalable lazy-greedy algorithm to select compact, high-coverage, low-redundancy subsets for both SFT and RAG. Experiments on MATH500 (SFT) and large-scale medical & history RAG corpora show that EDS subsets consistently outperform random, human-curated, and strong automated baselines (SuperFiltering, Qurating, Graph Entropy, etc.) in downstream performance and sample efficiency.

THEORETICAL CONCERNS:

1.How KCE can be used in RAG

KCE removes knowledge-level redundancy and keeps only the most informative, diverse passages, it directly improves the quality of retrieved context, making it highly suitable for RAG systems that rely on accurate and efficient retrieval.

2.Requires access to the full corpus

Added Monte Carlo approximation of KCE with rigorous theoretical guarantees (Appendix B.2)

3.Appears similar to clustering-based methods

Explicitly differentiated from embedding-clustering and token-level methods; emphasized our information-theoretic, knowledge-centric formulation with detailed comparison.

4.Evaluation fairness of RAG experiment

The knowledge set K is automatically derived from raw data and marked as intrinsic property for each information. Our algorithm used K to select a more dense, even distribution subset. For evaluation, the K will act as ground truth for the cosin similarity retrieve process.

ABLATION STUDY, FEW EXPERIMENTS:

1.Knowledge extractor fairness & concept noise (case sensitivity, duplicates, etc.)

New ablation studies (Appendix B.1) using different extractors and performance drop < 1.2%, proving strong robustness.

2.Experiments too limited (only MATH500, few models/datasets)

Adden GSM8K and MedQA-CoT-LLaMA31 data sets, as well as three different models in ablation study. We are now carrying out experiments on NuminaMath-CoT (900K), openbookqa (5K) and  PsiloQA (63K) datasets to see our algorithm performance on domain SFT works

3.No ablation studies / shallow analysis

Added ablations on frequency threshold, weighting scheme, extractor quality in Appendix B.1.

4.Experiments on 1M or bigger datasets

We plan to further evaluate our method on the NuminaMath-CoT dataset (900K) to demonstrate the robustness and generality of our results. The reviewer mentioned a prior study claiming that random selection may outperform data selection methods on very large datasets. However, their comparison was limited to token-level selection algorithms and did not consider the alignment between the distribution of the selected data and that of the test or evaluation sets. We believe our knowledge centric approach will achieve superior performance even on very large corpora.

PAPER STRUCTURE:

1.Matrix B construction hidden in appendix

Entire construction process of matrix B and KCE has been moved to the main paper; the manuscript is now fully self-contained.

2.Confusing figures / overstated claims

Merged redundant figures, fixed resolution, corrected bolding convention, and toned down all conclusions.

In summary, KCE and EDS offer a principled, knowledge-centric, and scalable framework for high-quality data selection. By explicitly optimizing semantic-level coverage and diversity, EDS consistently outperforms existing methods in both SFT and RAG settings, demonstrating its effectiveness for domain-specific SFT data selection as well as retrieval-based tasks.

---

### Note · Authors · 2026-01-26

I have read and agree with the venue's withdrawal policy on behalf of myself and my co-authors.

---

### Meta-Review · Area_Chair_wP4y · 2026-01-05

**Summary:**

This paper proposes a new approach for selecting an informative subset of examples for supervised fine-tuning (SFT) and retrieval-augmented generation (RAG). The key idea is to select examples that cover columns of a knowledge matrix that is automatically generated using an LLM. The proposed approach is a heuristic without a full analysis. The approach is evaluated on both SFT and RAG tasks, and compared to multiple baselines. The main strength of this paper is that the proposed algorithm is simple, and applies to both SFT and RAG. The reviewers had many concerns, which indicates that the paper needs a major revision and another round of reviews. The concerns, including mine, are:

* **No connection to the SFT or RAG objective:** The authors state their algorithm but never connect it to the SFT or RAG objective. Without such a connection, the algorithm is just a heuristic for selecting diverse examples that cover columns of an LLM-generated matrix. Such a connection can be made. Specifically, SFT is a log-likelihood maximization problem and the point of selecting diverse examples is to approximate the log-likelihood in all directions as well as if all data were available. I strongly suggest that the authors read [FisherSFT: Data-Efficient Supervised Fine-Tuning of Language Models Using Information Gain](https://proceedings.mlr.press/v267/deb25a.html). Their algorithm FisherSFT is greedy and comes with theoretical guarantees because log-determinant maximization is a submodular maximization problem. This is closely related, both in algebra and algorithmically, to the reviewed paper.

* **Algorithm design:** Key components of the algorithm, such as building the knowledge matrix, are hidden in Appendix. The construction of the matrix is not ablated. The matrix or parts of it are never validated by humans.

* **Limited experiments:** The experiments are limited to a single language model and only one dataset for SFT and two datasets for RAG. The experimental results would be more convincing with additional models and datasets.

* **Insufficient baselines:** The proposed approach is not compared to many baselines that seem relevant. Because the approach is coverage-based and maximizes diversity, diversity- and clustering-based baselines are relevant. This includes random sampling. I would also expect a comparison to Ask-LLM, which asks the LLM if the example should be included. This baseline is LLM-based but does not construct the knowledge matrix. See [From Selection to Generation: A Survey of LLM-based Active Learning](https://aclanthology.org/2025.acl-long.708/) for more potential baselines.

I do not recommend accepting this paper because it is a heuristic, which is not positioned properly with respect to prior works and compared to them.

**Reviewer Concerns:**

**Limited experiments** were addressed. **Algorithm design** was partially addressed. **Insufficient baselines** and **no connection to the SFT or RAG objective** were not addressed.

**Reviewer Scores:**

The score of Reviewer axhY is already 8 and would not go higher. The concerns of the other reviewers, mostly focused on **algorithm design** and **insufficient baselines**, were only partially addressed. Therefore, these reviewers would be unlikely to change their scores. Reviewer 3yBS stated this in their response.

---

### Decision · Program_Chairs · 2026-01-26

Reject